



# Integrated Observation of an Asymmetric Eddy Dipole in the South China Sea

Shuang Long[1], Fenglin Tian[1,2,3,*], Junwu Tang[2], Fangjie Yu[1,3], Fang Zhang[4], Wei Ma[5], Xinglong Zhang[3], Ge Chen[1,2,3,*]

[1] College of Marine Technology, Faculty of Information Science and Engineering, Ocean University of China, Qingdao, 266100, China
   [2] Frontiers Science Center for Deep Ocean Multispheres and Earth System, Ocean University of China, Qingdao, 266100, China
   [3] Laoshan Laboratory, Qingdao, 266237, China
[4] Key Laboratory of Marine Ecology and Environmental Sciences, Institute of Oceanology, Chinese Academy of Sciences, Qingdao, 266071, China
   [5] School of Mechanical Engineering, Tianjin University, Tianjin, 300350, China

*Correspondence to*: Ge Chen (gechen@ouc.edu.cn) and Fenglin Tian (tianfenglin@ouc.edu.cn)

**Abstract.** Mesoscale dipoles consist of mesoscale eddies with opposite signs. They are globally discovered and play a
significant role in eddy-eddy interactions. Due to the complexities of the marine environment, many dipoles are asymmetric, characterized by unequal strengths between the dipole eddies. The strong interaction has been observed between asymmetric dipole eddies, a phenomenon referred to as the "gear-like" process. Specifically, stronger dipole eddies generally drive weaker ones to move around, resulting in a reduction of discrepancies in their kinematic properties, such as rotational speed, amplitude, and eddy kinetic energy. An integrated observation of an asymmetric eddy dipole was conducted in the South China Sea in
April 2023. The general characteristics of the dipole eddies were derived from satellite altimeter data, and the evolution of their vertical structures was studied based on the joint data from Argo floats, gliders, drifters, and a survey vessel. Employing rigorous criteria, a 10−day successive coupling process of an asymmetric dipole was identified between a weaker anticyclonic eddy (AE) and a stronger cyclonic eddy (CE) from 13 to 22 April 2023. AE was initially weaker than CE in early April and strengthened when it coupled with CE, which is similar to the "gear-like" process. In addition, the drifting speed of the drifters
further confirmed the "gear-like" process between the target asymmetric dipole. The vertical temperature anomalies were surprisingly positive (~0.5 ºC at 50−300 m) on the CE periphery and reveal a distinct conical AE structure at 60−350 m. AE induced a significant sinking of dissolved oxygen saturation during the coupling process. Furthermore, the coupling interaction increased CE's positive temperature anomaly near the contact zone and deepened AE's temperature and dissolved oxygen saturation structures. Both thermohaline and biological responses provide evidence that the interaction between the asymmetric
dipole eddies impacted the vertical transport of water. These findings, on the one hand, support the observations that weaker dipole eddies strengthen after coupling with stronger ones. On the other hand, the results offer valuable vertical structure information of the asymmetric eddy dipole.



## 1 Introduction

Oceanic eddy dipoles, characterized by two contra-rotating eddies separated by a central jet (Fedorov and Ginsburg, 1989; Ni
et al., 2020), have garnered increasing attention due to their immense significance in eddy-eddy interactions (Chen et al., 2023;
Long et al., 2024). The advancement of satellite remote sensing technology, particularly the advent of satellite altimetry in the
late 20th century, has provided observational evidence for theoretical, experimental, and numerical studies of eddy dipoles.
For instance, Ahlnäs et al. (1987) identified seventeen dipole eddies around the Alaska Coastal Current using Landsat thematic
mapper data, highlighting their thermal signatures and significant contributions to cross-shelf mixing processes. Fedorov and
Ginsburg (1989) illustrated the peculiarities and formation of mushroom-like eddy dipoles around the Oyashio based on the
images from the Meteor-30 satellite. Cresswell et al. (2014) revealed a chain of cool and warm dipole vortices in the Great
Australian Bight through SeaWiFS and AVHRR images. Furthermore, Hughes and Miller (2017) reported nine specific
instances of fast-moving dipole eddy pairs (modons) based on satellite altimeter data, corroborating the theoretical predictions
and laboratory findings that modons propagated at a much higher speed than the Rossby wave speed on a rotating sphere (Stern,
1975; Haines and Marshall, 1987). Ni et al. (2020) and Long et al. (2024) revealed the universal and nonlinear composite
structure of eddy dipoles, as initially observed by Couder and Basdevant (1986), which plays a crucial role in their evolution.
Chen et al. (2023) discovered that the "gear coupling effect" of dipole eddies partly accounted for the mechanism and pattern
of the complex eddy propagations. Nevertheless, the satellite remote sensing data primarily captures the sea surface
characteristics and dynamics of eddy dipoles.

Compared to satellite retrievals, incidental in situ observations obtained from moorings, autonomous profilers, and shipboard
surveys supply valuable underwater information regarding eddy dipoles. By integrating the satellite altimeter data with
measurements from a lowered ADCP (Acoustic Doppler Current Profiler) and CTD (Conductivity-Temperature-Depth)
measurements, De Ruijter et al. (2003) detected an eddy dipole with a deep-reaching central jet that velocities exceeded over
20 cm/s at 2000 m depth, and assessed its capacity for water mass transport. A strong interaction was highlighted between a
mesoscale eddy dipole and a seamount located in the northern part of the Madagascar Ridge during a cruise (Vianello et al.,
2020). L'hégaret et al. (2014) investigated the collision, formation, and interaction of dipole eddies in the Gulf of Cadiz via a
unique joint data set including CTD, XBT (Expendable Bathythermograph), Rafos floats, satellite altimeter data, and sea
surface temperature (SST) data. Through a substantial number of quality-controlled Argo temperature and salinity (T/S)
profiles, Ni et al. (2020) constructed a global composite vertical structure of the dipoles, and Yang et al. (2023) endeavored to
derive a universal law for the temporal evolution of eddy dipoles interacting with cold filaments in the Southern Indian Ocean.
Despite the increasing oceanographic cruises and Argo Real-time Ocean Observing Network in the worldwide ocean, there
remains a significant gap in the comprehensive observation of dipole eddies that account for approximately 30%−40% of the
total eddies (Ni et al., 2020).

Dipole is asymmetric when it consists of eddies with unequal strengths (Long et al., 2024; Couder and Basdevant, 1986).
Dipolar eddies moved along a straight (curved) line when the dipoles were symmetric (asymmetric) owing to the self-





propelling mechanism (Couder and Basdevant, 1986; Fedorov et al., 1989; Kubryakov et al., 2024). Recent studies have indicated that the weaker CE may obtain energy from the stronger AE, thereby enhancing its strength (Kubryakov et al., 2024; Li et al., 2020). Belonenko et al. (2021) documented an uneven vortex pair in the Loften and found that the smaller cyclone rotated around the large anticyclone in a manner analogous to the Moon around the Earth. Furthermore, Long et al. (2024)

analysed the evolution of asymmetric dipole eddies based on the satellite altimeter data for the period January 1993–December 2020. Their findings demonstrated that the stronger dipole eddies generally drove the weaker ones to revolve due to the self-propelling mechanism, resulting in a reduction of discrepancies in the kinematic properties of the eddies (e.g., rotational speed, amplitude, eddy kinetic energy, and so on) after the asymmetric dipoles destructed. This coupling effect was referred to as the "gear-like" process, wherein the stronger dipole eddies acted as the driver gear while the weaker ones were akin to the driven

gear. Therefore, asymmetric dipoles and the interaction between the eddies deserve more in situ observations.

South China Sea (SCS), the largest semi-enclosed marginal sea in the North Pacific Ocean, is an ideal region for surveying and studying eddy dipoles owing to the abundant energetic mesoscale eddies (Huo et al., 2024; Chen et al., 2011; He et al., 2018). Eddy dipoles have been observed in the SCS (Liao et al., 2021; Wang et al., 2021; Chu et al., 2017; Zhang et al., 2013). Dipole structures with a cyclonic eddy (CE) to the north and an anticyclonic eddy (AE) to the south occurred frequently off

the Vietnam coast (Chu et al., 2017; Wang et al., 2006). Their formation was closely linked to the monsoon and was influenced by the El Niño/Southern Oscillation (ENSO) (Chu et al., 2017; Chu et al., 2014). Liao et al. (2021) investigated the photosynthesis-irradiance response to such an eddy dipole. Additionally, numerous eddy dipoles were generated in the western Luzon Strait and subsequently propagated southwestward (Wang et al., 2021; Zhang et al., 2013; Lin et al., 2016; Sheu et al., 2010). Their generation was occasionally relevant to the Kuroshio intrusion, thereby modulating the water/heat/salinity

transport and affecting the circulation in the SCS (Wang et al., 2021; Zhang et al., 2013; Sheu et al., 2010). Nevertheless, the studies and in situ observations paid less attention on the coupling process between the asymmetric dipole eddies in the SCS. As a consequence, this work conducted a collaborative observation experiment of an asymmetric dipole in the SCS, including satellite remote sensing platforms, shipborne survey equipments, and autonomous mobile floats. Comprehensive datasets and the methodology for dipole detection are outlined in Sect. 2. Sect. 3 elaborates on the evolution of the target asymmetric dipole

from satellite remote sensing platforms. Sect. 4 analyses the observations from shipborne survey equipments, and autonomous mobile floats. Finally, Sect. 5 presents the concluding remarks.

## 2 Data and Methods

### 2.1 Altimeter data and eddy dipole detection

The satellite altimeter sea level anomaly (SLA) product, SEALEVEL_GLO_PHY_L4_NRT_008_046

(https://marine.copernicus.eu/), is derived from the Copernicus Marine Environment Monitoring Service (Cmems, 2025). The gridded data is calculated with respect to a twenty-year mean from 1993 to 2012. It is estimated by Optimal Interpolation, merging the L3 along-track measurements from various altimeter missions available. This product offers a temporal resolution



of one day and a spatial resolution of 1/4° by 1/4°. It is updated daily and serves in near-real-time applications, making it suitable for the pre-planning of ship survey routes.

Eddy was identified based on the SLA geometry (Tian et al., 2020; Liu et al., 2016). A Gaussian filter with the half-power wavelength cutoffs of 20° in longitude by 10° in latitude was applied to the global SLA map before extracting contours. Iterating the closed contours with no more than one local SLA extreme from the outermost inward to the innermost, the eddy boundary was defined as the contour with the maximum average geostrophic current speed, and the eddy center was designated as the centroid of the innermost contour (Fig.1(a)). Eddy was then tracked based on the minimum dimensionless similarity parameter

between two consecutive time steps (Tian et al., 2020). For each eddy identified at time step $t_1$, the next candidates were the eddies identified at time step $t_2$ that lay within a circle with a 0.5° radius from its centroid. Then dimensionless similarity parameter of each candidate was calculated as follows:

$$S_{t_1,t_2} = \sqrt{\left(\frac{\Delta d}{d_0}\right)^2 + \left(\frac{\Delta a}{a_0}\right)^2 + \left(\frac{\Delta A}{A_0}\right)^2 + \left(\frac{\Delta E}{E_0}\right)^2}$$

where $\Delta d$ represented the separation distance between the eddy centers at time steps $t_1$ and $t_2$, and $\Delta a, \Delta A,$ and $\Delta E$ were the

differences in amplitude, area, and eddy kinetic energy (EKE) between the eddies at time steps $t_1$ and $t_2$, respectively. $d_0, a_0, A_0,$ and $E_0$ were characteristic constants. The next eddy was assigned when $S_{t_1,t_2}$ was the minimum.

The coupling process of the eddy dipole was extracted through two steps following a previous algorithm (Long et al., 2024). First, a pair of AE and CE was identified as a dipole structure when the distance between two eddy centers was shorter than the sum of their radii (eddy radius refers to the radius of the area-equal circle of the eddy boundary (black circles in Fig.1 (a)).

Second, the coupling process was defined as the segments in the AE and CE trajectories that AE and CE were identified as dipole structures.

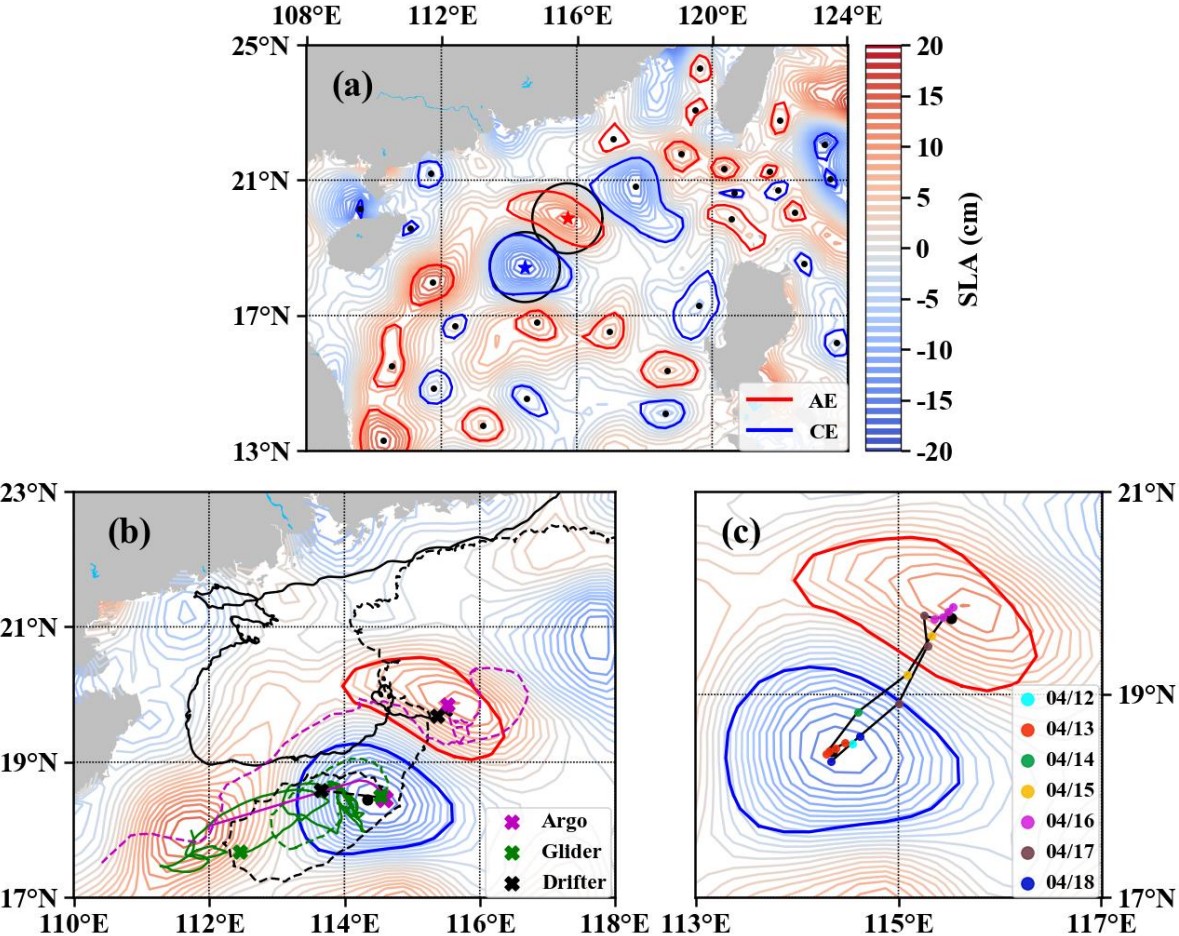

**Figure 1: The integrated observation of an asymmetric dipole in the SCS. (a) The eddy distribution in SCS on 13 April 2023, (b) the placements and trajectories of two Argo floats (purple crosses), two gliders (green crosses), and two drifters (black crosses), and (c) seventeen survey stations (colorful dots) and route (black line) of the survey vessel crossing dipole eddies. The red (blue) curves represent AEs (CEs), and the black dots are the eddy centers. The red and blue stars in (a) represent the observed dipole eddies, and the black circles correspond to the area-equal circles of the eddy boundary.**

## 2.2 The field observations

Eddies in the SCS were detected and monitored for several months in advance utilizing near-real-time SLA data. Ultimately, an eddy pair in the western Luzon Strait was selected for observation (Fig.1 and Fig.2). During the research cruise taking place from April 10 to April 25, 2023, the position of the target eddy pair was first detected each day following the method in Sect. 2.1, and its next position was forecasted using two different algorithms (Ge et al., 2023; Chen et al., 2024). The results were then communicated to the survey vessel to guide the cruise and float deployment.

This research cruise was designed to gain the hydrographic and biological sections crossing the eddy centers of the CE-AE pair using a RBRmaestro3 multi-channel logger and an Underwater Vision Profiler (UVP). The survey vessel was underway



from Zhanjiang, Guangdong Province on 10 April 2023 and successfully reached the target CE after two days (12 April 2023). Subsequently, the vessel arrived at the AE center on 15 April 2023 and turned back to CE on 17 April 2023. The vessel exited the target CE on 18 April 2023 and returned to Zhanjiang on 25 April 2023. Throughout the cruise, a total of twenty-nine stations were surveyed, with seventeen consecutive stations traversing the dipole eddy centers from April 12 to April 18, 2023

(colorful dots in Fig.1(c)). RBRmaestro3 multi-channel logger supports up to ten sensors on a single platform, allowing for a diversity of sensor configurations that can be fine-tuned for various applications. Its substantial storage capacity and reliable battery power enable extended deployments with higher sampling rates, making it particularly suitable for in situ observations. UVP serves as a low-power, cost-effective, and deep-ocean-rated in situ camera designed for automatically monitoring particles and plankton from autonomous platforms. It is capable of counting and sizing large particles with an equivalent

spherical diameter (ESD) exceeding 100 $\mu m$. At each station, the RBRmaestro3 multi-channel instrument collected data of conductivity, temperature, salinity, oxygen, chlorophyll a concentration (Chl a), and so on from the ocean surface to 350 m depth at a 5 m interval, while UVP captured images of plankton and marine snow at the same depth.

Drifter is an extremely valuable tool for reflecting near-surface ocean currents (Lumpkin et al., 2013). It is widely used to investigate the coherent mesoscale eddies that can trap Lagrangian particles over long periods of time (Lumpkin, 2016; Liu et

al., 2016; Chen et al., 2021). To observe the geostrophic currents around the target eddies, two drifters were deployed in the CE and AE on April 12[th] and April 16[th], 2023, respectively (two black crosses in Fig.1(b)). Drifter_1 recorded data from April 12[th] to 24[th] July and stayed within the CE until May 5[th], while Drifter_2 ran for 101 days from April 16[th] to August 3[rd] and was trapped in the AE until May 11[th].

The HM2000 Argo profile float is novel observation equipment and is able to automatically drift over a long time. After

approbation by the International Argo Project, it has been utilized for the construction and maintenance of the Global Argo Real-time Ocean Observing Network (Zhang, 2018). Two hydrographic Argo floats were employed to daily record the T/S profiles inside the target AE and CE, with a maximum measure depth of 2000 m (two purple crosses in Fig.1(b)). One Argo float (No.898133) was trapped within the CE and recorded a total of 12 profiles from 12 to 16 April 2023. The other Argo float (No.897875) was deployed inside the AE on 16 April 2023 and finally ceased operation on 17 September 2023. It

remained within the AE for only 8 days as the AE weakened.

The Petrel underwater glider, engineered by Tianjin University, is a buoyancy-driven and propeller-driven unmanned underwater vehicle that can continuously measure physical parameters (e.g., temperature and salinity) for a long period (Yang et al., 2019). In this work, Gilder_1 was deployed inside the CE on 12 April 2023, and Glider_2 was placed at the southwest outside of the CE on the same day (two green crosses in Fig.1(b)). As scheduled, the gliders should have traversed the dipole

eddies to acquire high-resolution measurements from the surface down to 1000 m. However, dipole eddies coupled strongly, leading to the high-speed eddy current between them. As a consequence, the gliders lacked sufficient power to traverse the target dipole eddies like the cruise and ultimately remained within the dipole CE for a long time.





**2.3 Other datasets**

The National Oceanic and Atmospheric Administration (NOAA) provides three marine products for this work. The NOAA
National Centers for Environmental Information (NCEI) is responsible for the World Ocean Atlas 2023 (WOA23) and the
ETOPO (Earth Topography) Global Relief Model (ETOPO 2022) (accessible at https://www.ncei.noaa.gov/).

WOA23 is a collection of objectively interpolated mean fields for oceanographic variables at 102 standard depth levels for the
World Ocean, including the temperature, salinity, oxygen, phosphate, silicate, and nitrate derived from profile data from the
World Ocean Database (Ncei, 2022). It consists of annual, seasonal, and monthly mean fields on grid resolutions of 5°, 1°, and
1/4°. For this work, the WOA23 dataset was used to characterize the typical northern SCS water and Kuroshio water and
calculate the T/S anomalies from the observations of Argo floats and gliders.

ETOPO 2022 is a full-coverage, seamless, gridded topographic and bathymetric bare-earth elevation dataset that integrates
topography, bathymetry, and shoreline data from regional and global datasets. It is available in Ice Surface and Bedrock
versions at 15-, 30-, and 60-arc-second resolutions. This work applied the Bedrock version at 15-arc-second resolution to
analyze bathymetry in the SCS.

The high-resolution SST product is from the NOAA Physical Sciences Laboratory (PSL) Optimum Interpolation Sea Surface
Temperature Version 2 (OISST V2, available at https://psl.noaa.gov/data/gridded/data.noaa.oisst.v2.html) (Huang et al., 2021).
It incorporates multi-platform observations (satellites, ships, buoys, and Argo floats) into a regular global grid. The dataset is
interpolated to fill gaps on the grid and create a spatially complete map of SST. To mitigate discrepancies arising from
differences in platforms and sensor biases, satellite and ship observations are calibrated against buoy data. The temporal and
special resolutions of OISST V2 is a day and 1/4°, respectively. It was utilized to analyzed the SST fronts in the SCS in April,
2023

**3 Observations based on satellite remote sensing data**

**3.1 Development of the target eddy dipole**

185    Figure 2(a) depicts the trajectories and origins of the target eddies. CE was generated approximately at 118.0º E, 18.5º N and
remained active for 142 days from February 8th to June 29th, 2023, and AE originated near 117.5º E, 22.0º N and survived for
85 days from February 28th to May 23rd, 2023. After generation, CE moved northwestward while AE traveled southeastward.
Subsequently, the target AE merged with another smaller AE and strengthened. AE then encountered the target CE on 3 March
2023 and they generally propagated southwestward along the isobaths, with AE following the 1000 m isobath and CE adhering
to the 3000 m isobath. It can be attributed to the alignment of the isolines of potential vorticity (PV=$f$/H, where $f$ and H are
the planetary vorticity and water depth, respectively) with the bathymetric contours (Zhang et al., 2013).

Figure 2(b) shows the ratio of the relative vorticity ($\zeta$) to planetary vorticity ($f$) at each grid point, from the time when that grid
point was closest to a dipole eddy center (Hughes and Miller, 2017). Its absolute value, known as the Rossby number $|\zeta/f|$,

typically ranges from 0.1 to 0.4. As expected, the negative (positive) relative vorticity along the CE (AE) trajectory in the
Northern Hemisphere is continuous. Notably, the relative vorticity "trace" caused by the target mesoscale eddies is surrounded
with a continuous "coat" of relative vorticity of opposite signs, which may be link to their movement together over a long time
(Couder and Basdevant, 1986). In contrast, the relative vorticity away from the eddy trajectories shows typical amplitudes with
a relatively random distribution  (Hughes and Miller, 2017).

Based on the T/S profiles from Argo floats and WOA23 climatological data, a T-S (temperature-salinity) diagram is plotted in
Fig. 2(c) to distinguish the water properties within the target eddies. Numerous studies have revealed that the northern SCS
water is cooler and fresher compared to the typical Kuroshio water in the upper layer (< 300 m), and the mesoscale eddies
generated in the western Luzon Strait play a significant role in the process of Kuroshio Intrusion. The T-S plots in Fig. 2(c)
shows that the water mass within the target eddies is evidently similar to the northern SCS water compared to the typical
Kuroshio water.

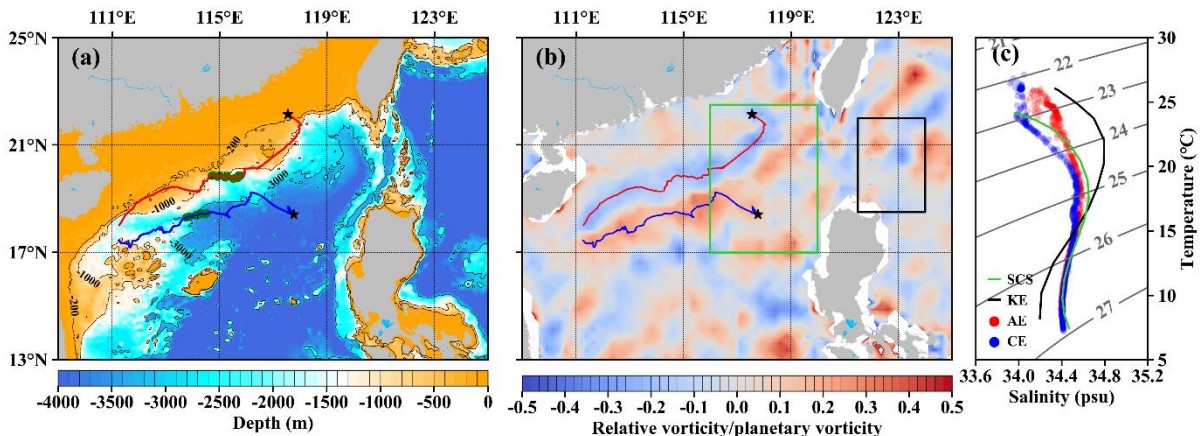


**Figure 2: (a) Bathymetry of the northeastern SCS. The 200, 1000, and 3000 m isobaths are marked with thin black lines. (b) The vorticity maps that the relative vorticity is divided by planetary vorticity at each grid point. The red and blue lines denote the trajectories of AE and CE revealed in this study, respectively. The black stars represent the eddy origins. (c) The T-S diagrams of Argo profiles within the dipole eddies. The green and black lines are the averaged T-S curves for the northeastern SCS (green box,**
**116 ºE–120 ºE, 17 ºN–22.5 ºN) and Kuroshio area (black box, 121.5 ºE–124 ºE, 18.5 ºN–22 ºN), respectively, based on World Ocean Atlas 2023 climatological data.**

During the 85−day coexistence period spanning February 28th − May 23rd, 2023, the pair of AE and CE was recognized as
dipole structures from 13 to 22 April, from 11 to 12 May, and from 19 to 22 May 2023 in terms of the aforementioned strict
criteria in Sect. 2.1. This work focuses on the first continuous coupling process lasting for 10 days. Table 1 lists the mean
values of eddy properties throughout their entire lifetime and during their 10−day continuous coupling process. As previously
described (Long et al., 2024; Yu et al., 2023), both eddies become stronger during the continuous coupling process, and the
dipole CE is more active than the dipole AE since AE was located in shallower water.




**Table 1. Primary surface properties of AE and CE during the whole lifetime and 10−day coupling process.**

| Properties | Rotational speed (cm/s) whole/coupling | Amplitude (cm) whole/coupling | EKE (cm$^2$/s$^2$) whole/coupling | Radius (km) whole/coupling |
|---|---|---|---|---|
| AE | 21.6/27.6 | 4.6/6.9 | 175.3/261.8 | 77.7/97.4 |
| CE | 31.0/33.0 | 8.0/9.8 | 365.7/373.9 | 94.9/113.8 |

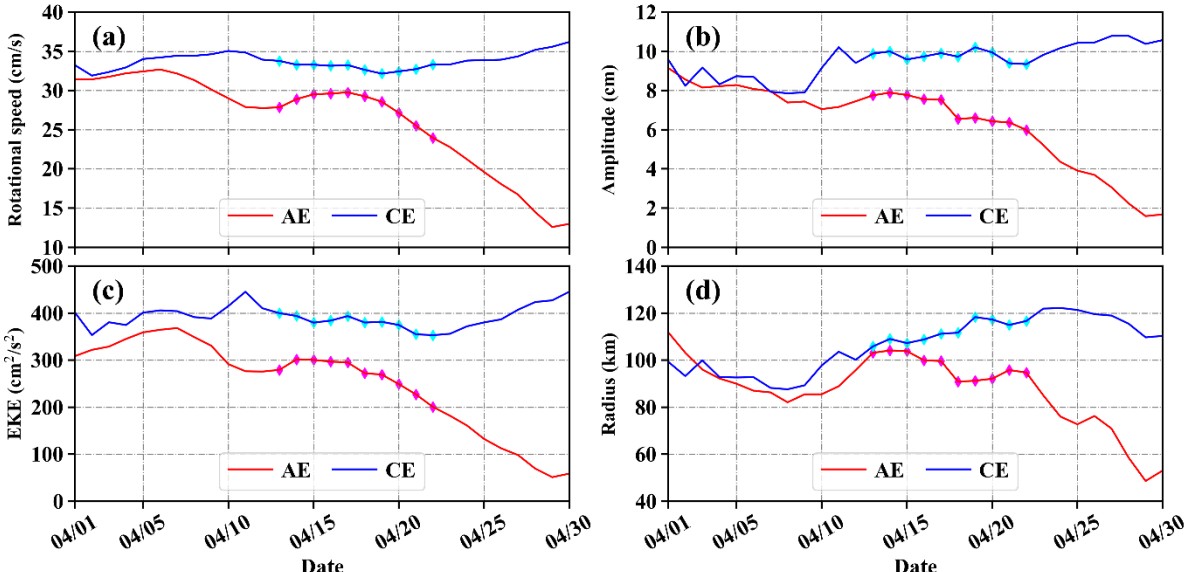

**Figure 3: The temporal evolution of (a) rotational speed, (b) amplitude, (c) EKE, and (d) radius for the dipole AE and CE. The diamonds mark the continuous coupling process from 13 to 22 April.**

To explore the interaction between asymmetric dipole eddies in detail, Figure 3 displays the evolution of eddy kinematic properties in April. In early April, both AE and CE exhibited roughly equal rotational speed (around 32.0 cm/s), amplitude (approximately 8.5 cm), eddy kinetic energy (EKE, around 340.0 cm$^2$/s$^2$), and radius (approximately 100.0 km). On the following days from April 6th to 10th, AE became weaker, with a reduction in rotational speed (down by ~5.0 cm/s), amplitude (down by ~1.0 cm), EKE (down by ~100.0 cm²/s²), and radius (down by ~10.0 km), while CE exhibited a slight increase in

amplitude (up by ~2.5 cm), radius (up by ~14.0 km), and EKE (up by ~40.0 cm$^2$/s$^2$). It is worth noting that AE subsequently strengthened owing to the coupling effect that the weaker AE was propelled by the powerful CE, which is referred to as "gear-like" process between asymmetric dipole eddies by Long et al. (2024). This coupling effect was observed between an asymmetric eddy dipole that was composed of a mesoscale AE and a submesoscale CE (Kubryakov et al., 2024). The stronger AE lost a large part of energy to the weaker submesoscale CE because of the coupling effect. More specifically, after AE

coupled with CE to form a dipole structure, AE experienced a modest increment in rotational speed (up by ~2.5 cm/s), amplitude (up by ~1.0 cm/s), radius (up by ~ 20.0 km), EKE (up by ~25.0 cm$^2$/s$^2$) from April 10th to 15th. Despite the coupling





effort, CE was ultimately unsuccessful in preventing AE from weakening, especially after the separation of the dipole structure. AE rapidly weakened from April 15th, and by April 30th, its rotational speed, amplitude, and EKE significantly declined to ~12.2 cm/s, ~1.4 cm, ~49.6 cm$^2$/s$^2$, and ~48.4 km, respectively. Eventually, AE completely dissipated after May 23rd, while
CE continued moving southwest along the isobath until it disappeared after June 22nd.

**3.2 Kinematic properties of the asymmetric dipole**

Figure 4 depicts the spatial distribution of SLA, speed of surface geostrophic currents, and the corresponding acceleration of the dipole eddies. The magnitude of SLA induced by dipole CE was approximately −14.0 cm in April, while the values caused by AE exhibited a similar pattern of rotational speed in Fig. 3(a). That is to say, the SLA at the AE core first increased and
then reduced after it coupled with CE. The eddy current between the dipole eddies flowed faster, reaching a peak of approximately 50.0 cm/s on April 14th, which is larger than the global mean value of approximately 40.0 cm/s (Long et al., 2024; Ni et al., 2020). Although AE and CE were not identified as a dipole structure from April 23rd due to the strict distance constraint, the eddy current speed remained at approximately 40.0 cm/s. This is the reason why two gliders failed to traverse the target CE and AE. It is noteworthy that the corresponding acceleration around AE and CE on April 13th was evident.
Moreover, the positive and negative acceleration values were alternately distributed along the line through their eddy centers, indicating the spatially non-uniform variation in speed around the dipole eddies, as previously observed by Long et al. (2024). The difference, however, is that the positive (negative) acceleration during the dipole formation did not become negative (positive) during the dipole destruction.

Affected by eddy advection, the drifting direction and speed of drifters reflect the eddy rotational direction and speed, resulting
loopers in their trajectories. (Liu et al., 2016; Lumpkin, 2016; Chen et al., 2021). Therefore, this work utilized two drifters to investigate the coupling effect between the dipole eddies. The Video S1 presents the daily distribution of the geostrophic currents around the dipole eddies and the moving speed of drifters, and four snapshots are displayed in Fig. 5. A smoothing was applied to remove high-frequency signals and noise. Clearly, Drifter_1 drifted counterclockwise along a larger circular trajectory, whereas Drifter_2 moved clockwise and there was a small looper in its trajectory. This is link to eddy intensity. The
counterclockwise (clockwise) movement of Drifter_1 (Drifter_2) was consist with the rotation of CE (AE) in the Northern Hemisphere. Drifter_1 and Drifter_2 were deployed at the eddy center of the target AE and CE on 12 and 16 April 2023, respectively (Fig. 5 (a) and (b)). The drifting speed increased when Drifter_1 and Drifter_2 moved from eddy core to boundary. Nevertheless, the drifting speed failed to manifest the coupling effect well since Drifter_1 and Drifter_2 were not under the influence of the jets between the dipole eddies during the 10−day coupling process (Fig. 5 (c)). In comparison, Drifter_2 drifted
fast at around 70 cm/s on 9 May when it was affected by the jets, evidencing the coupling effect between the dipole eddies.



**Figure 4: SLA (left panels), speed of surface geostrophic currents (middle panels), and corresponding acceleration (right panels) in the northern SCS in April 2023. The solid red (blue) curves represent the target AEs (CEs), and the black dots are the eddy centers. The dashed red (blue) curves are other AEs (CEs).**

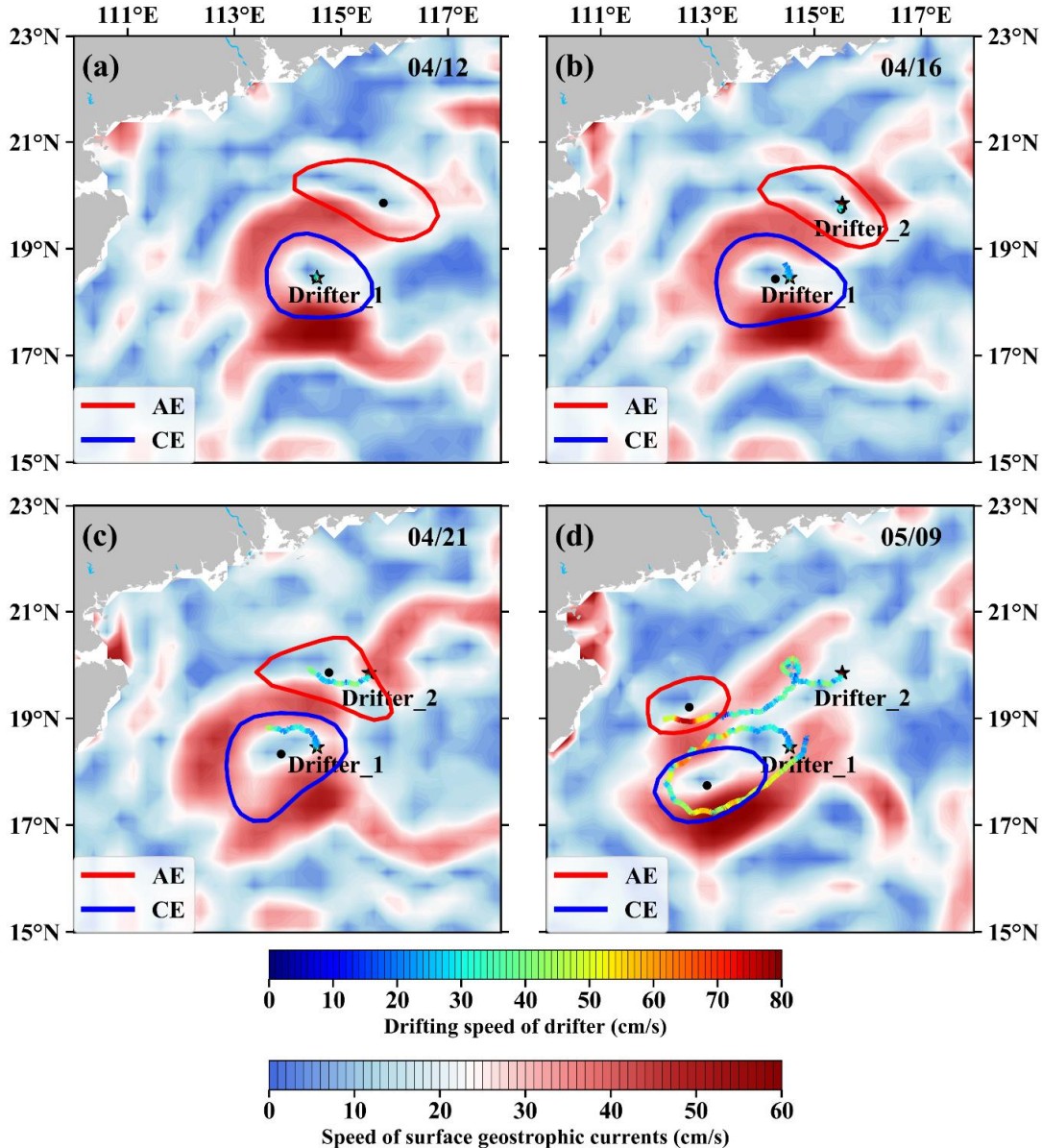

**Figure 5: The speed of surface geostrophic currents and the drifting speed of drifters until (a) 12 April 2023, (b) 16 April 2023, (c) 21 April 2023, and (d) 9 May 2023. The solid red (blue) curves represent the observed AE (CE), the black dots are the eddy centers, and the black stars are the origins of drifters. The Video S1 presents the daily distribution of the speed of surface geostrophic currents and the drifting speed of drifters.**

## 3.3 Eddy shape of the asymmetric dipole

To quantify the eddy deformation observed in Fig. 4, shape error and scale ratio are calculated in the dipole coordinate (Fig. 6). The shape error quantifies the area discrepancy between the eddy boundary and its corresponding fitted circle (Mason et





al., 2014). Denoting two diameters as $D_1$ and $D_2$ (Fig. 6(a)), scale ratio is defined as the ratio of $D_2$ to $D_1$. Long et al. (2024) discovered that the weaker dipole eddies were more prone to deformation. Likewise, the AE studied in this work (Fig. 6(a)) is

more deformed than the CE, as evidenced by the greater shape error (approximately 30% to 50%) and scale ratio (greater than 1) for AE compared to CE (shape error: 20% to 30% and scale ratio: almost less than 1). Concretely, $D_1$ and $D_2$ of CE varies almost simultaneously so that the scale ratio remains close to 1. In comparison, $D_1$ of AE varies slightly while $D_2$ first increases and then falls rapidly, leading to a significant variation in the scale ratio.

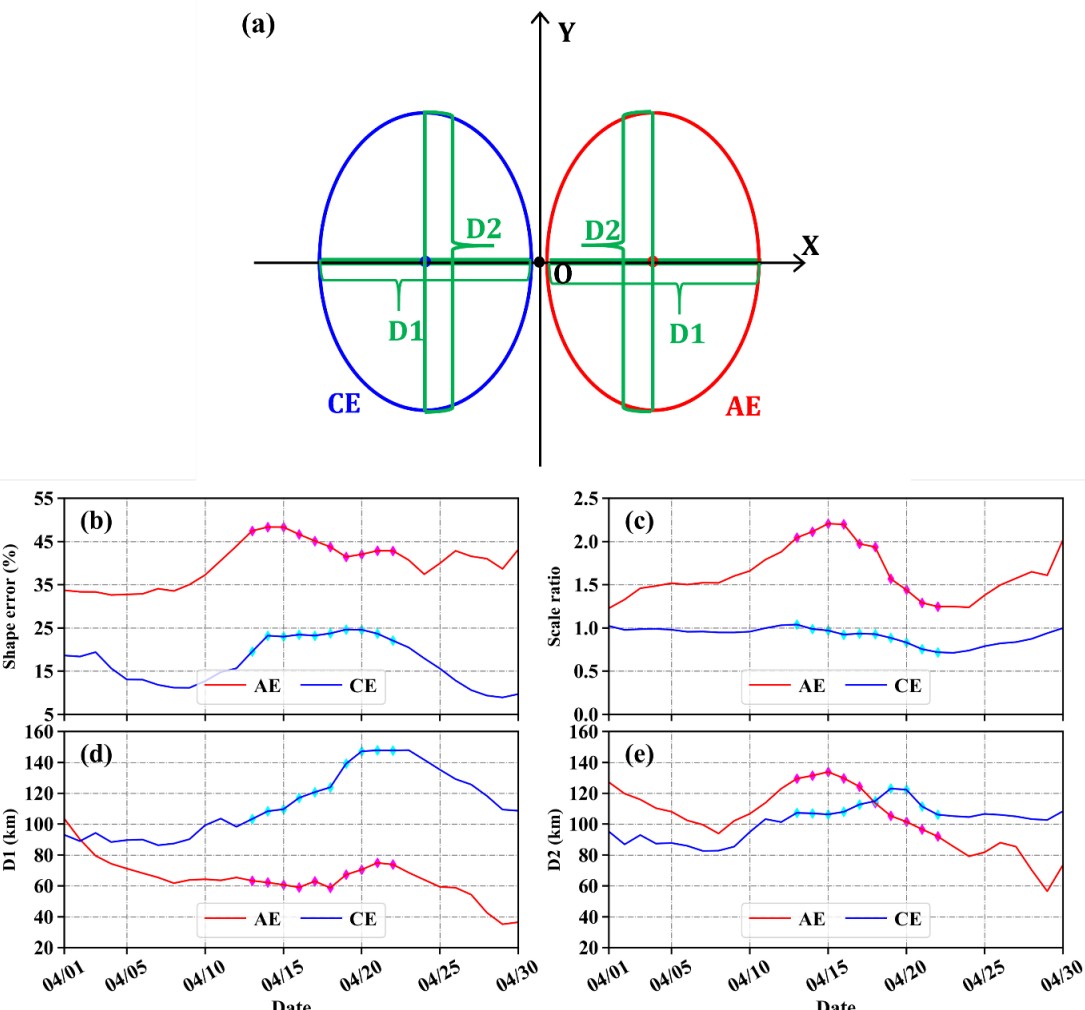

**Figure 6: (a) A diagram of dipole coordinate. The midpoint of the two dipole eddy centers is designated as the coordinate origin $O$ of the dipole-centric coordinate system, with the CE center and AE center located on the negative and positive X-axis, respectively. The temporal evolution of (b) shape error, (c) $D_1$, (d) $D_2$, and (e) scale ratio for the dipole AE and CE. The diamonds mark the continuous coupling process.**

### 3.4 Influence of eddy dipole on SST front

Oceanic fronts are prevalent throughout the whole ocean and are defined as narrow transition zones between water masses with distinctly different properties (Ma et al., 2023; Qiu et al., 2020). The horizontal gradients of SST are frequently utilized to delineate SST fronts (Qiu et al., 2017; Ma et al., 2024). SST fronts often occur in proximity to mesoscale eddies and are influenced by their dynamics. Ma et al. (2023) documented a vertical SST front between a pair of AE and CE in the Kuroshio Extension. Consequently, Figure 7 displays the SST gradients around the target dipole eddies in April. The SST gradient was

calculated as follows: $G = \sqrt{G_x^2 + G_y^2}$, where the $G_x$ and $G_y$ are the zonal and meridional SST gradient components, respectively (Qiu et al., 2017). SST fronts are detected when the gradients are greater than 0.05 °C/5km (dashed black lines in Fig. 7).

The SST gradients within the dipole eddies are a little weak. Prior to April 19th, the SST front in the CE was situated in the southwest. Subsequently, it underwent a counterclockwise rotation under the influence of eddy advection and subsequently

merged with the SST front around the AE on April 25th. The SST front in the CE strengthened from April 13th to April 17th but weakened in the following days. In contrast, the SST front around the AE first weakened before April 19th, accompanied by a reduction in the corresponding area. After April 19th, the front expanded in area, especially merging with the SST front inside the CE.

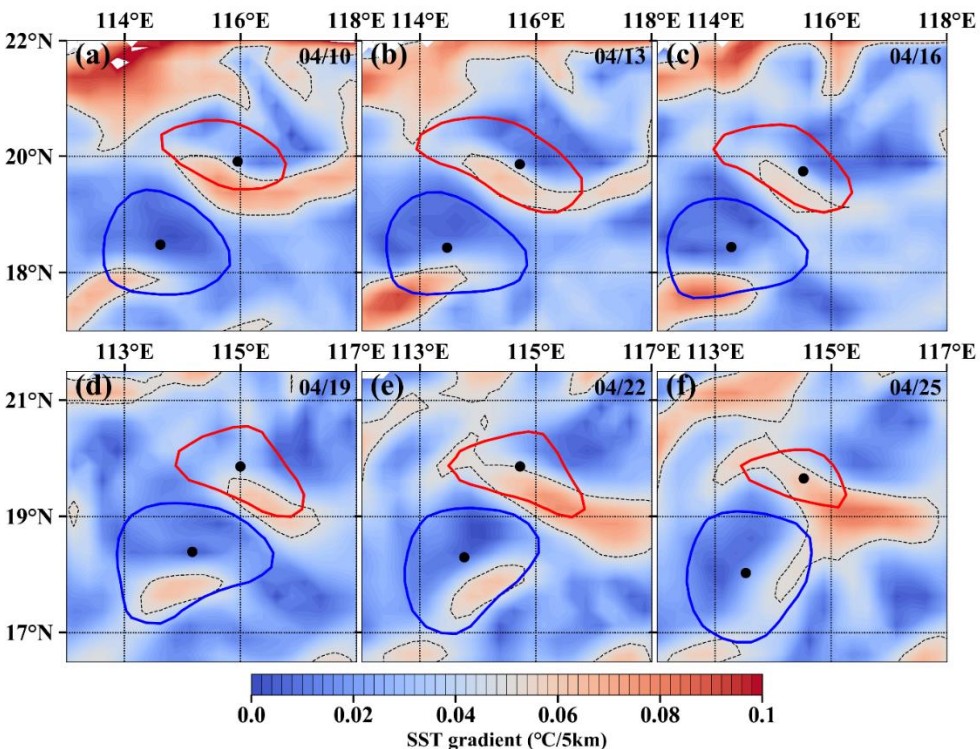

**Figure 7: The distribution of SST gradients in the northern SCS in April 2023. The red (blue) curves represent AEs (CEs), and the black dots are the eddy centers. The dashed black curves are the 0.05 °C/5km contours.**



# 4 Vertical structures based on the profile floats and survey vessel

## 4.1 Eddy-following profiles

The thermohaline structures of dipole eddies are investigated based on the T/S profiles from Argo floats (Fig. 8 and
Supplementary Material Video S2). Argo No.898133 obtained the T/S profiles of the CE eddy core during April 12th −16th, and Argo No.897875 gained the T/S profiles from the AE eddy core to its boundary as dipole movement during April 16th −22nd (Supplementary Material Video S2). In Fig. 8, the black contours represent T/S values, while the colored shading indicates T/S anomalies. The thermocline at the CE eddy core evolved smoothly during the observed period. The composite of temperature anomaly in the CE is predominantly negative, with an approximate value of -1.5° in the SCS (Zhang et al.,
2024; He et al., 2018). However, the target dipole CE induced small negative (approximately -0.6 °C) at eddy core, but marginally positive temperature anomalies (~0.5 °C) at roughly 50−300 m depth not far from eddy core. This distribution may be associated with the water parcel during the CE formation. As the trajectory in Fig. 2 depicted, CE was generated in the relatively southern SCS on 8 February 2023, at a time when the temperature in the southern area exceeded that in the northern region. Salinity anomaly within the CE is negative below 50 m, peaking at 50−100 m with a value of -0.15 psu. The positive
temperature anomaly observed at 100−200 m in Fig. 8 manifests a typical AE structure as expected (He et al., 2018; Chaigneau et al., 2011). The temperature anomaly in the AE closely resembles the composite of the three-dimensional structure of temperature anomaly (He et al., 2018), which gradually weakens from eddy core to boundary. Additionally, surface salinity experienced a noticeable decline from approximately 34.30 psu to 34.0 psu from 16 to 23 April, with a distinct downward trend in the contours after April 21. The subsurface salinity maximum, recorded at 34.60 psu, was located at approximately
150 m depth.



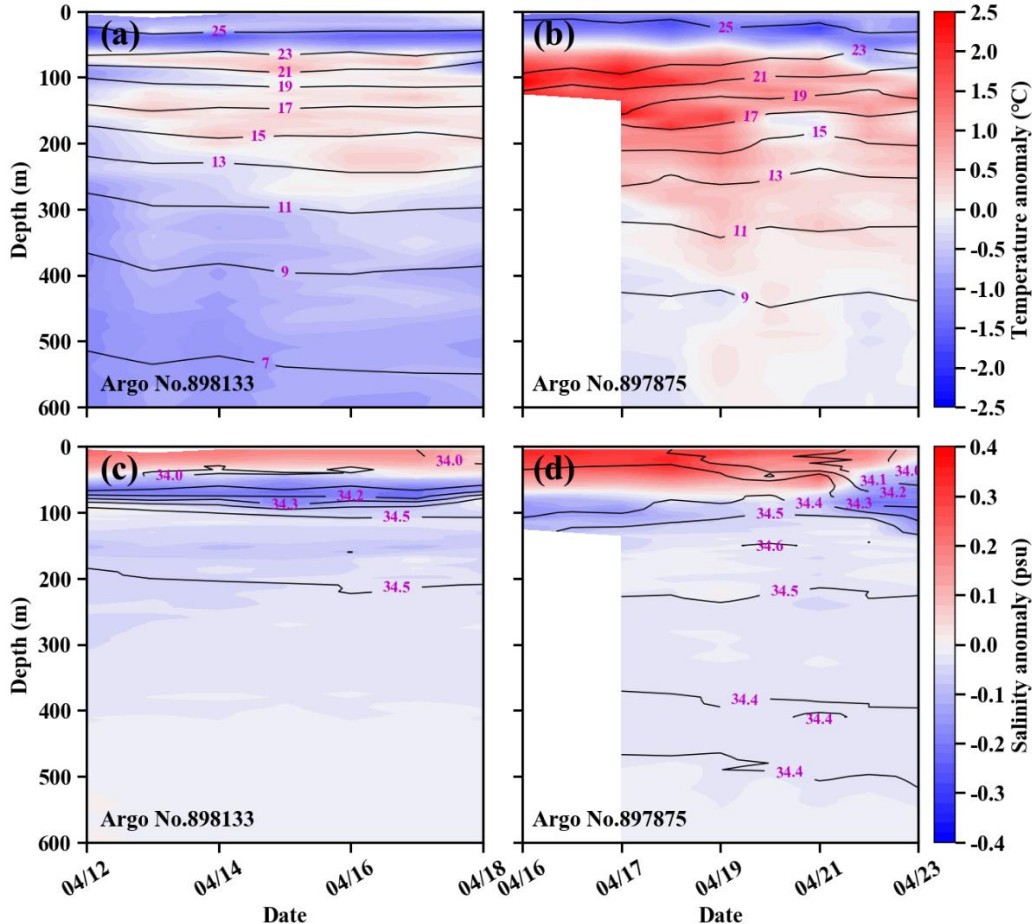

**Figure 8: T/S (black lines) and corresponding anomaly (colored shadow) observed by the Argo floats. Blanks in each plot indicate the absence of data. The Video S2 presents the positions of Argo floats and the daily distribution of T/S profiles in detail.**

Although two gliders failed to traverse the CE and AE, they provided a large number of high-resolution T/S profiles of CE

(Fig. 9). To assess the relative position of the gliders to the dipole CE, the distance and azimuth angle were calculated (Fig. 9(a)). The distance is the spatial separation between the CE center and the gliders. The azimuth angle $\theta$ is defined as the angle of the ray from the CE center to the glider relative to the ray through dipole eddy centers, and it ranges from −180 ° to 180 °. The smaller the absolute angle, the larger the distance, indicating that the position of the glider is closer to the contact zone between the dipole CE and AE.

Glider_1 stayed within the CE during the whole observation period and moved anticlockwise (Supplementary Material Video S3). Similar to the Argo No.898133 observation, the temperature anomaly is slightly positive at roughly 50−300 m depth and the salinity anomaly is almost positive above 50 m. Furthermore, the bottom of the positive temperature anomaly apparently deepened when Glider_1 was located at the CE center from April 28[th] to May 1[st].

After the launch, Glider_2 moved northeastward and entered the CE on April 17th (Supplementary Material Video S3). It
reached the CE center on April 21st, left the CE on April 29th, and almost remained at the contact zone between the dipole AE
and CE until May 3rd. Apparently, the T/S anomalies recorded by Glider_2 from April 17th to 21st were positive between 50 m
and 300 m in the southwestern part of the CE. When Glider_2 stayed at the contact zone after April 26th, the deeper red shades
implied a more pronounced positive temperature anomaly, and the deeper blue shades signified a more significant negative
salinity anomaly. This observation indicates that the T/S anomalies closer to the contact zone are evident because of the
interaction between the dipole eddies.

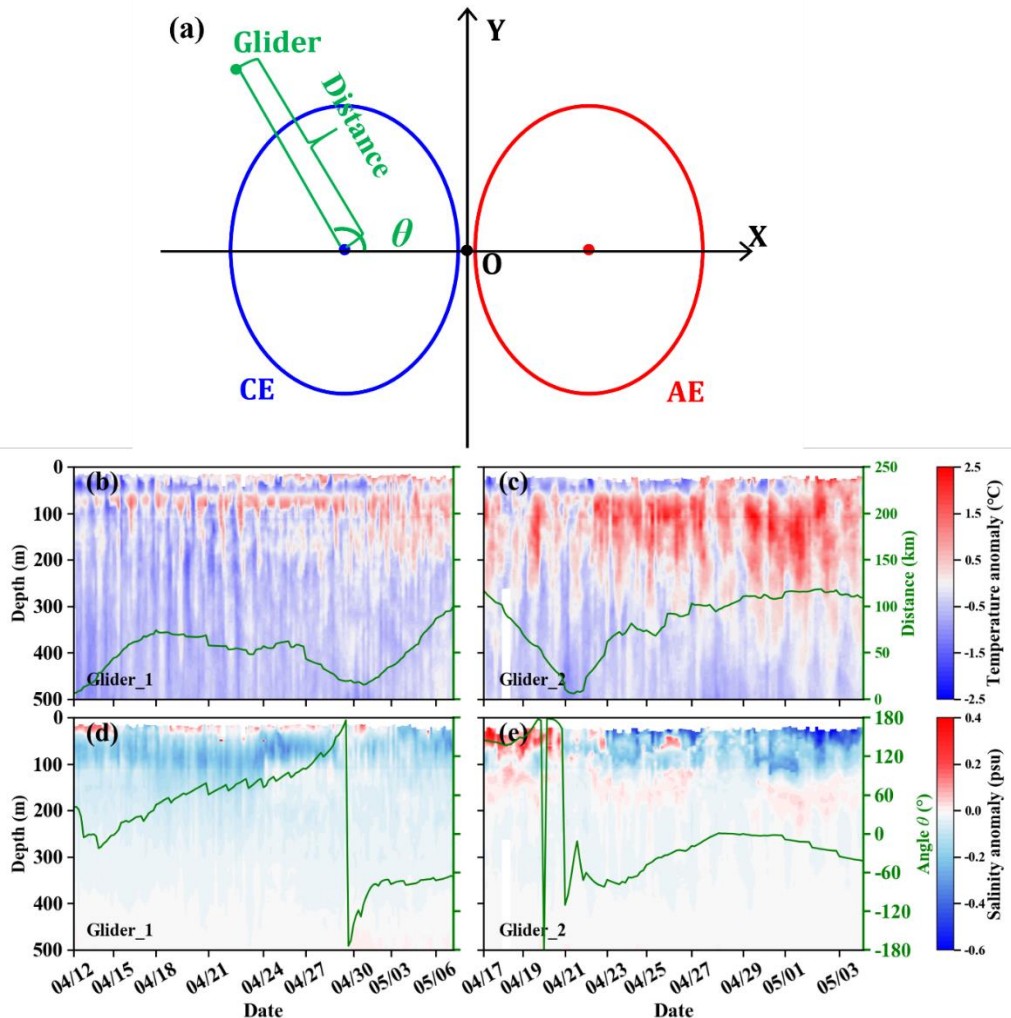

**Figure 9:** T/S (black lines) and corresponding anomaly (colored shadow) observed by the gliders. Blanks in each plot indicate the
absence of data. (a) A diagram of dipole coordinate. The distance is the spatial separation between the CE center and the gliders.
The azimuth angle θ is defined as the angle of the ray from the CE center to the glider relative to the ray through dipole eddy centers.
The Video S3 presents the glider positions and the daily distribution of T/S profiles in detail.



## 4.2 Sections crossing dipole eddies

The RBRmaestro3 supplies the vertical thermohaline profiles between the dipole eddies from April 12th to 18th (Figure 10 and Fig. 1(c)). The black thermohaline contours illustrate that CE caused upwelling while AE induced downwelling, leading to sharp changes in T/S at the contact zone between the dipole AE and CE. As monitored by Argo floats and gliders, thermoclines

inside the CE evolved smoothly, whereas temperature anomaly exhibited positive temperature anomalies (~0.5°C) from 50 m to 250 m near the CE periphery, alongside a negative salinity anomaly (~0.15 psu) from 50 m to 100 m. Furthermore, the positive temperature anomaly at around 150 m within the CE increased from 0.4 °C to 0.6 °C from April 12th to 18th, suggesting possible heat exchange between the dipole CE and AE. The temperature anomaly characterizes a distinct conical AE structure at 60−350 m depth. The downwelling induced by the AE generated a maximum temperature anomaly of 3.2 °C at

approximately 150 m, which is higher than the mean value of 1.5 °C centered at 90 m in the SCS (He et al., 2018) but is lower than the values caused by the AEs associated with the Kuroshio intrusion into the SCS (Zhang et al., 2013; Wang et al., 2021). Besides, the isotherms at 12−24 °C and the isohalines at 34.5 psu within the AE deepened on April 17th compared to April 16th, indicating that the coupling effect between dipole eddies contributed to the vertical water transport.

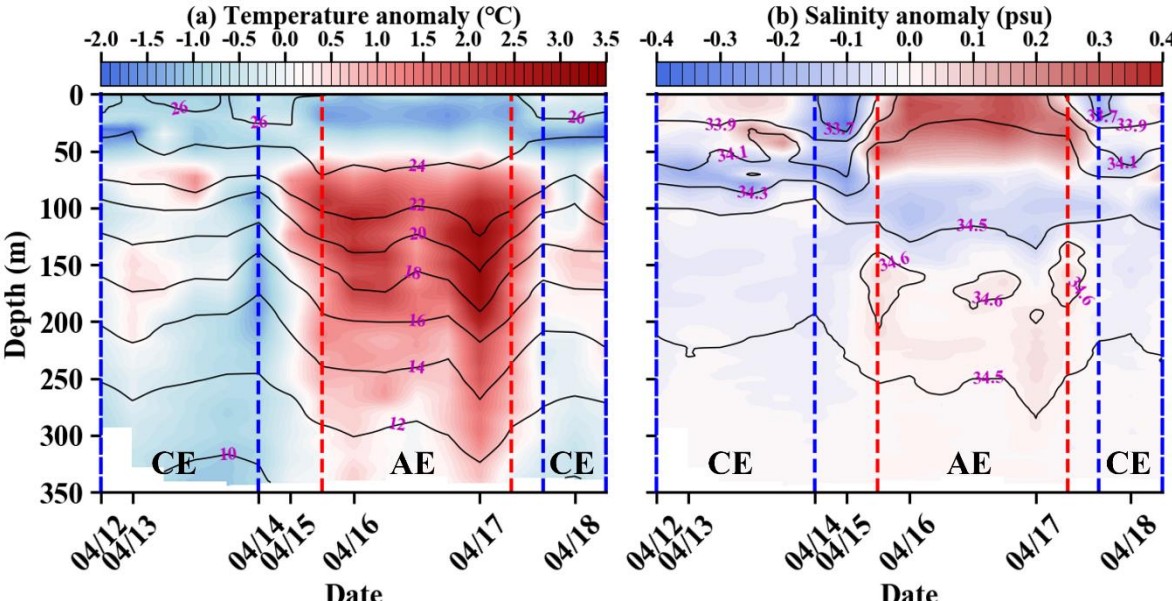

**Figure 10: (a) Temperature (black lines) and corresponding temperature anomaly (colored shadow), and (b) salinity (black lines) and corresponding salinity anomaly (colored shadow) crossing the dipole eddy centers. The profiles between two dashed blue (red) lines indicate stations are located within the CE (AE), and the profiles between a dashed red line and a dashed blue line indicate stations are located at the contact zone. Blanks in each plot indicate the absence of data.**

Figure 11 exhibits the sections of Chl *a* concentration and dissolved oxygen (O₂) saturation between the dipole CE and AE

gained by RBRmaestro3. Numerous studies have revealed that Chl *a* concentration at the subsurface is significantly greater than that at the surface, a common phenomenon known as subsurface chlorophyll *a* maximum (SCM) in the worldwide ocean

(Chen et al., 2022; Perry et al., 2008). Consistent with prior observations (Chen et al., 2022), the SCM is prominently observed at 50−100 m, with the Chl $a$ concentration exceeding 0.8 $\mu g/L$. Notably, two peaks in Chl $a$ concentration were recorded: one at the CE core on April 13$^{th}$ with a value of around 4.0 $\mu g/L$ (it is noted that the 2.0 $\mu g/L$ of color bar includes all values
exceeding 2.0 $\mu g/L$) and the other at the CE boundary on April 17$^{th}$ with a value of 1.8 $\mu g/L$. Furthermore, the bottom of SCM in the CE and AE is generally situated at the 63% and 80% contours (dashed gray lines in Fig. 11(a)) of dissolved $O_2$ saturation, respectively. Below the euphotic zone, the dissolved $O_2$ saturation within the CE substantially declined to 60%, whereas the 60% dissolved $O_2$ saturation contour sank to around 250 m in the AE. This observation further supports the notion that CE induces water mass upwelling while AE leads to water mass downwelling. Additionally, the oxygen-rich surface
waters on April 17$^{th}$ were observed to descend to greater depths compared to that on April 15$^{th}$, resembling the variations in thermoclines in Fig. 8. The variations further evidence that the coupling effect between dipole eddies impacted the vertical transport of waters.

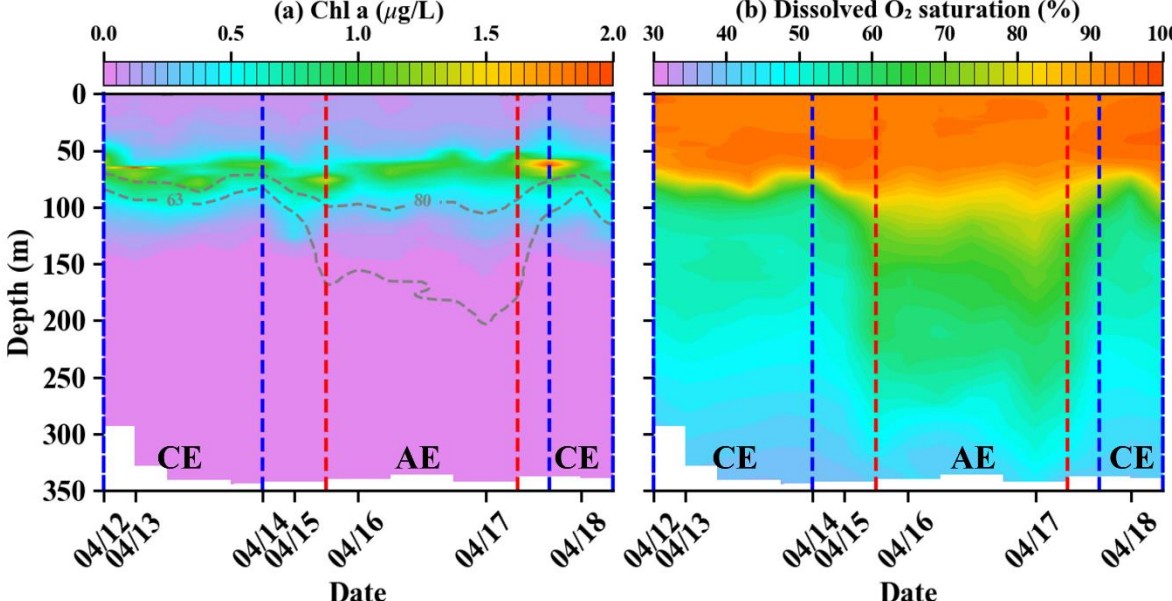

**Figure 11: (a) Chl $a$ concentration and (b) dissolved O₂ saturation crossing the dipole eddy centers. The dashed gray lines in (a) are**
**the dissolved oxygen (O₂) saturation. The dashed red and blue lines have the same meaning as Fig.8. Blanks in each plot indicate the absence of data. It is noted that the 2.0 μg/L of the color bar is the value exceeding 2.0 μg/L.**

Equipped on the RBRmaestro3, UVP offers the vertical sections of particle concentrations between the dipole eddies (Fig. 12). The various size particles were roughly divided into three classes following the previous study (Scherer et al., 2011). Microorganisms are the particles with an ESD of 2−200 $\mu m$, small zooplankton have an ESD of 200−2000 $\mu m$, and the ESD
of large zooplankton exceeds 2000 $\mu m$. The particle concentrations in the CE are vaguely larger than those in the AE. Apparently, particles are predominantly found above 50 m, particularly the zooplankton are active at depths shallower than 25

m. The particle concentration of microorganisms is the largest with an order of $10^4$, the small zooplankton is the second one to have a concentration of an order of $10^2$, while the large zooplankton exhibits the lowest concentration, at an order of $10^1$.

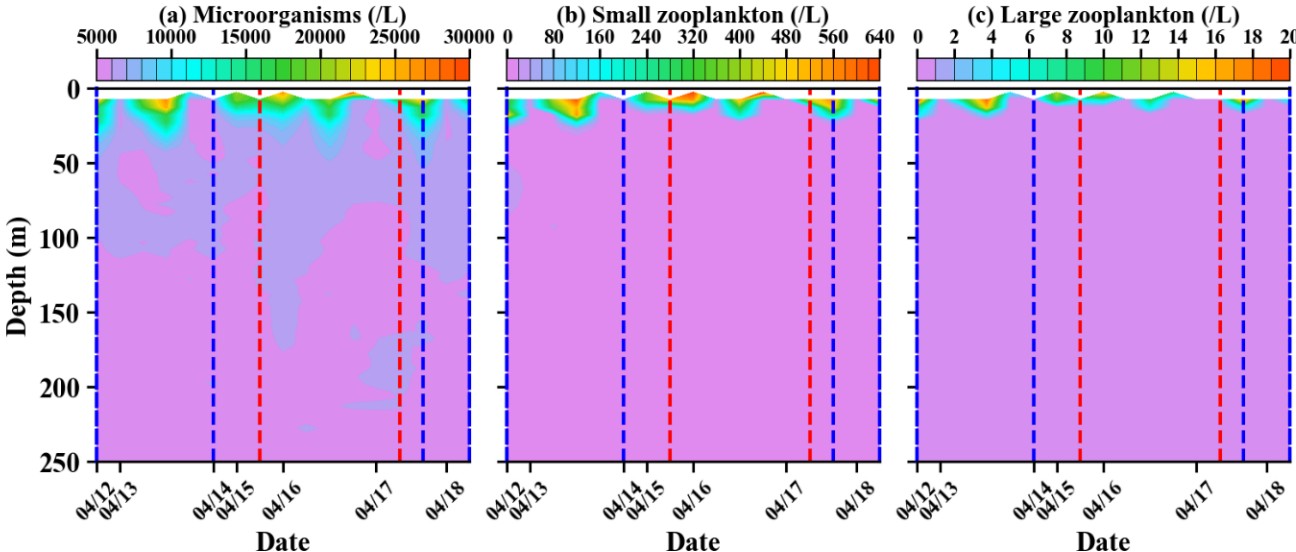

Figure 12: The particle concentrations of (a) microorganisms, (b) small zooplankton, and (c) large zooplankton crossing the dipole eddy centers. The dashed red and blue lines have the same meaning as Fig.8. Blanks in each plot indicate the absence of data.

## 5 Data availability

The dataset is available on the Zenodo repository under the DOI: https://doi.org/10.5281/zenodo.15122958 (Long et al., 2025).

## 6 Summary and discussion

Long et al. (2024) have discovered that the weaker dipole eddies enhance affected by the stronger dipole eddies due to the self-propelling mechanism, and this coupling effect referred to as the "gear-like" process. This work provides evidence of an asymmetric dipole in the SCS to further substantiate the coupling effect and investigates the evolution of their vertical structures. A pair of AE and CE were generated in the west of the Luzon Strait and subsequently propagated together towards southwest. A 10−day successive coupling process from 13 to 22 April 2023 was identified. The kinematic characteristics of dipole eddies reveal that although the stronger CE made the weaker AE strengthen as a result of the coupling effect, the CE failed to prevent AE from weakening. Additionally, the eddy current between the dipole eddies flowed too fast to make the gliders not traverse the dipole eddies.

A conical shape of temperature anomaly ranging from 60 m to 350 m was observed within the AE, with a maximum of 3.2 ℃ centered at around 150 m. Surprisingly, positive temperature anomalies (~0.5 ℃) at 50−300 m was found near the CE periphery. Chl $a$ concentration at 50−100 m exceeded 0.8 $\mu g/L$. The 63% and 80% contours of dissolved $O_2$ saturation roughly

corresponded to the SCM lower boundary within the CE and AE, respectively. Both thermohaline and biological responses demonstrate that the contours of temperature, salinity, and dissolved $O_2$ saturation inside the AE deepened during the latter half of the coupling process under the influence of the coupling effect. These observations illustrate that the interaction between dipole eddies influences the vertical water transport. In addition, SST fronts are affected by the dipole eddies.

The coupling process of oceanic dipole eddies occurs in a short and random time, leading to limited predictability and very few actual observations of dipoles in the ocean (Travkin et al., 2022). Although a comprehensive observation for an asymmetric eddy dipole has been conducted, the vertical profiles collected here lacked information regarding the eddy evolution before dipole formation.

**Author contributions**

JT, GC, FY, FT, FZ, WM, and XZ conceived and designed the integrated observation. All the authors performed the integrated observation. SL and TL wrote and edited the manuscript.

**Declaration of Generative AI in scientific writing**

During the preparation of this work, the authors used WORDVICEAI in order to improve readability and language, not to replace key researcher tasks such as interpreting data or drawing scientific conclusions. After using this tool/service, the authors

reviewed and edited the content as needed and took full responsibility for the content of the publication.

**Competing interests**

The contact author has declared that none of the authors has any competing interests.

**Disclaimer**

Publisher's note: Copernicus Publications remains neutral with regard to jurisdictional claims made in the text, published maps,

institutional affiliations, or any other geographical representation in this paper. While Copernicus Publications makes every effort to include appropriate place names, the final responsibility lies with the authors.

**Acknowledgements**

The authors are grateful to the reviewers for their work and constructive comments. The altimeter data was freely downloaded from the CMEMS website (https://marine.copernicus.eu/). The WOA23 and the ETOPO 2022 were freely downloaded from



NOAA NCEI website (https://www.ncei.noaa.gov/). The high-resolution SST product was freely downloaded from the NOAA

PSL website (https://psl.noaa.gov/data/gridded/data.noaa.oisst.v2.html).

**Financial support**

This work was jointly supported by the Science and Technology Innovation Project for Laoshan Laboratory (Grant Nos. LSKJ202201406 and LSKJ202204303), and National Natural Science Foundation of China (Grant No. 42030406). We are

very grateful to the anonymous Reviewers for the constructive comments and helpful suggestions which significantly help us to improve the quality of this paper.

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
