# Peer review of "Integrated Observation of an Asymmetric Eddy Dipole in the South China Sea"

_Earth System Science Data, 2025_

## Author Comment (AC1)

<Journal Name> Earth System Science Data

< MS No.> essd-2025-276

<MS type> Data description paper

<Manuscript Title> Integrated Observation of an Asymmetric Eddy Dipole in the South China Sea

**Dear Editors and Reviewers,**

We highly appreciate the detailed and valuable comments of the referees on our manuscript entitled "Integrated Observation of an Asymmetric Eddy Dipole in the South China Sea (MS No. essd-2025-276)". These comments are all valuable and helpful for revising and improving our paper, as well as providing important guidance for our research. In the past few days, we have referred to the comments and improved the paper.

As follows, we would like to clarify some of the points raised by the Associate Editor and Reviewers. The original comments begin with "**Questions and Comments**" and are quoted in normal font, the replies are in blue letters, the revised sentences and phrases are in red letters, and the line number in the revised manuscript is highlighted in yellow. We appreciate the Editors/Reviewers' warm work and taking the time to review the manuscript, and we hope that the corrections will meet with approval.

Yours Sincerely,

Shuang Long, Fenglin Tian*, Junwu Tang, Fangjie Yu, Fang Zhang, Wei Ma, Xinglong Zhang, Ge Chen[*]

2025-12-01

**1. Questions and Comments:** More observations on eddy dipoles and the results are very important for knowing their 3D structures and the couplings (or interactions) between them. In this paper, Argo, gliders, drifters, and research vessel were used to observe an eddy dipole which may be associated with Kuroshio intrusion in the South China Sea, the relevant data obtained in this cruise are presented, and a gear-like process associated with the dipole is suggested. However, the integrated observation scheme proposed in this paper is not very effective, at least, the argos, gliders and so on didn't provide very significant data, and accordingly it is not sure that "the gear-like process" is appropriate for the couplings for eddy dipole. Maybe there is no gearing effect between AE and CE.

**Answer:** Thanks for your comments. Our previous work revealed the "gear-like" process of asymmetric dipole based on global analyses. Its distinctive feature is that stronger dipole eddies generally drive weaker ones to move around, resulting in a reduction of discrepancies in their kinematic properties via eddy-eddy interaction, such as rotational speed, amplitude, and eddy kinetic energy. Therefore, this manuscript first confirmed the observed dipole eddies exhibit "gear-like" process during their first coupling process according to their kinematic properties and eddy shape derived from satellite altimetry data (Section 3 in this manuscript). Subsequently, we further analyzed their evolution of vertical structure from floats and survey vessel observations and found that the vertical structure of the asymmetric dipole eddies responses to their changes in surface feature.

**2. Questions and Comments:** "anomalies" presented in this paper are not the original data. How they are derived should be clarified. Why are there blank in Argo data (No. 897875) on 16-April? And, accordingly, will the anomalies in temperature and salinity be altered?

**Answer:** Thank you for pointing this out. The text regarding "anomalies" was added in Line 175-180 as "This work utilized the annual statistics of WOA23 dataset with a grid size of $1/4°$ to characterize the typical northern SCS water and Kuroshio water and calculate the T/S anomalies from the observations of Argo floats, gliders, and RBRmaestro3. The annual and observed T/S profiles were first resampled at the same depth through linear interpolation. The T/S anomalies were then obtained by subtracting annual T/S profiles from the observed T/S profiles." Therefore, the "blank" Argo data (No. 897875) does not influence the T/S anomalies. The Argo was artificially set to hover at about 120 m depth.

**3. Questions and Comments:** The RBRmaestro3 seems to provide more reliable data for the eddy dipole presented. The salinity anomaly in the transitional zone between AE and CE gives pronouncing signal of interaction between AE and CE, however, there is no adequate support for "gearing" process. The same is with the Chl-a.

**Answer:** Thanks for your comments. The "gear-like" process of asymmetric dipole was proposed based on the eddy surface feature. Therefore, this manuscript first confirmed the observed dipole eddies exhibit "gear-like" process during their first coupling process according to their kinematic properties and eddy shape derived from satellite altimetry data (Section 3 in this manuscript). The profiles from floats and survey vessel observations provide pronouncing signal of eddy-eddy interaction during the 10-day coupling process, especially the deeper contours of T/S profiles and dissolved $O_2$ saturation sections on 17 April responses to the increase in eddy kinematic properties on 16-17 April in Fig.3. This indicates the response between changes in surface feature of dipole eddies and changes in vertical structure via eddy-eddy interaction.

[Figure]

**Figure 3: The temporal evolution of (a) rotational speed, (b) amplitude, (c) EKE, and (d) radius for the dipole AE and CE. The diamonds mark the continuous coupling process from 13 to 22 April.**

[Figure]

**Figure10: (a) Temperature (black lines) and corresponding temperature anomaly (colored shadow), and (b) salinity (black lines) and corresponding salinity anomaly (colored shadow) crossing the dipole eddy centers. The profiles between two dashed blue (red) lines indicate stations are located within the CE (AE), and the profiles between a dashed red line and a dashed blue line indicate stations are located at the contact zone. The gray in each plot indicates missing values.**

[Figure]

**Figure11: (a) Chl *a* concentration and (b) dissolved O₂ saturation crossing the dipole eddy centers. The dashed gray lines in (a) are the dissolved oxygen (O₂) saturation. The dashed red and blue lines have the same meaning as Fig.8. The gray in each plot indicates missing values. It is noted that the 2.0 μg/L of the color bar is the value exceeding 2.0 μg/L.**

**4. Questions and Comments:** Other indexes by microorganisms and zooplanktons as shown by Fig.12, give no signal of AE and CE. More observations from other cruises or from other published documents rather than a case, may be necessary.

**Answer:** Thanks for bringing this to our attention. We had updated the Fig.12 on a logarithmic scale. It now provides pronouncing signal regarding the diel vertical migration of zooplankton. The related text was added in Line as "Furthermore, the observed eddies are found to exert a significant influence on the vertical distribution of small zooplankton. A secondary subsurface maximum of small zooplankton occurs at approximately 75 m depth in close proximity to the CE core. In contrast, abundances are significantly suppressed within the AE core, especially at the 80–160 m depth, but exhibit elevated values within the AE–CE transition zone. This finding is consistent with Guidi et al. (2012) that concentrations of buoyant particles increase within the mesoscale frontal zone of AE-CE dipole." Please see Line 405-410 in the revised manuscript. Other research well demonstrated the eddy-eddy interaction on the diel vertical migration of zooplankton (Guidi et al., 2012.), and this manuscript more evidence to support the findings.

Guidi, L., Calil, P. H., Duhamel, S., Björkman, K. M., Doney, S. C., Jackson, G. A., ... & Karl, D. M. (2012). Does eddy-eddy interaction control surface phytoplankton distribution and carbon export in the North Pacific Subtropical Gyre?. Journal of Geophysical Research: Biogeosciences, 117(G2).

[Figure]

**Figure 12: The particle concentrations of (a) microorganisms, (b) small zooplankton, and (c) large zooplankton crossing the dipole eddy centers. The dashed red and blue lines have the same meaning as Fig.8. The gray in each plot indicates missing values. Blanks in (c) mean there is no large zooplankton.**

**5. Questions and Comments:** How does the Kuroshio intrusion cause AE, CE or eddy dipole?

**Answer:** Thanks for your comments. Nan et al. (2011) summarized three paths of Kuroshio intrusion through Luzon Strait: looping path, the leaking path, and the leaping path. Looping path is related to the Kuroshio Loop Current eddy shedding event. It results in high-salinity or high-temperature water. Based on the T-S plot from Argo floats and climatological WOA23 data (in Fig. 2(c)), the water mass within the target eddies is evidently similar to the northern SCS water compared to the typical Kuroshio water. Since this manuscript pays more attention to the evolution of coupling process, the impact of Kuroshio intrusion on the generation of dipole eddies will be further studies in the future work.

[Figure]

**Figure 2: (a) Bathymetry of the northeastern SCS. The 200, 1000, and 3000 m isobaths are marked with thin black lines. (b) The vorticity maps that the relative vorticity is divided by planetary vorticity at each grid point. The red and blue lines denote the trajectories of AE and CE revealed in this study, respectively. The black stars represent the eddy origins. (c) The T-S diagrams of Argo profiles within the dipole eddies. The green and black lines are the averaged T-S curves for the northeastern SCS (green box, 116 ºE–120 ºE, 17 ºN–22.5 ºN) and Kuroshio area (black box, 121.5 ºE–124 ºE, 18.5 ºN–22 ºN), respectively, based on World Ocean Atlas 2023 climatological data.**

Nan, F., Xue, H., Chai, F., Shi, L., Shi, M., and Guo, P.: Identification of different types of Kuroshio intrusion into the South China Sea, Ocean Dyn., 61, 1291-1304, 10.1007/s10236-011-0426-3, 2011.

---

## Author Comment (AC2)

<Journal Name> Earth System Science Data

< MS No.> essd-2025-276

<MS type> Data description paper

<Manuscript Title> Integrated Observation of an Asymmetric Eddy Dipole in the South China Sea

**Dear Editors and Reviewers,**

We highly appreciate the detailed and valuable comments of the referees on our manuscript entitled "Integrated Observation of an Asymmetric Eddy Dipole in the South China Sea (MS No. essd-2025-276)". These comments are all valuable and helpful for revising and improving our paper, as well as providing important guidance for our research. In the past few days, we have referred to the comments and improved the paper.

As follows, we would like to clarify some of the points raised by the Associate Editor and Reviewers. The original comments begin with "**Questions and Comments**" and are quoted in normal font, the replies are in blue letters, the revised sentences and phrases are in red letters, and the line number in the revised manuscript is highlighted in yellow. We appreciate the Editors/Reviewers' warm work and taking the time to review the manuscript, and we hope that the corrections will meet with approval.

Yours Sincerely,

Shuang Long, Fenglin Tian*, Junwu Tang, Fangjie Yu, Fang Zhang, Wei Ma, Xinglong Zhang, Ge Chen*

2025-12-01

**1. Questions and Comments:** This manuscript integrates multiple observations from satellite altimeters, Argo floats, gliders, drifters, and survey vessels to investigate a case of an asymmetric eddy dipole in the South China Sea, with a particular focus on the evolution of its vertical structure. However, two reasons prevent me from recommending it for further consideration in Earth System Science Data. **Answer:** Thanks for your comments.

**2. Questions and Comments:** (1) I cannot identify sufficient novel aspects of this work to match the high impact expected for ESSD. Although the authors provide a dataset collected from multiple observation platforms, it is limited to a single local case of an asymmetric eddy dipole. Therefore, I do not believe this dataset has the potential to attract sufficient interest from the international research community. As noted in the authors' review in the introduction, many previous studies have revealed the vertical structure of asymmetric eddy dipoles using observational data (Line 50-63). In addition, the results of this manuscript mainly offer a simple descriptive analysis of the observations, without presenting any new findings that distinguish this study from previous work. The "gear-like" process has already been proposed in their previous studies based on global analyses. The vertical structures of temperature, salinity, dissolved oxygen, chlorophyll, and zooplankton have already been well documented in numerous previous studies. Although I acknowledge that most of those studies focused on anticyclonic and cyclonic eddies rather than asymmetric eddy dipoles, it remains unclear what new findings this work provides that distinguish it from previous research using the same type of dataset. The authors claim that their observations provide evidence of the impacts of asymmetric eddy dipoles on the vertical transport of water. However, this point has already been well demonstrated by previous numerical and observational studies, such as Guidi et al. (2012).
**Answer:** Thanks for your comments. We totally agree that many previous studies analyzed the vertical structure or explored the impact of eddy-eddy interaction on ecosystem of asymmetric eddy dipoles. Our previous work revealed the "gear-like" process of asymmetric dipole on the global scale, but it is primarily based on the eddy surface feature. The underwater structure and corresponding evolution of asymmetric dipole on global scale are a challenging task due to a lack of enough underwater information. Therefore, more *in-situ* data regarding asymmetric dipole are needed, and this manuscript provides another typical coupling process in SCS. In addition, this manuscript focuses on the response between changes in surface feature of dipole eddies and changes in vertical structure via eddy-eddy interaction. Since CE keeps steady and AE changes a lot during the coupling process, we concentrated on AE's vertical structure and found it is influenced by vertical transport of water.

**3 Questions and Comments:** (2) As a manuscript submitted as a data description paper, I believe the current style and format do not provide sufficient details about the dataset. I recommend that the authors pay more attention to thoroughly describing the data collection methods, sensors used, processing procedures, temporal resolution, instructions, and application prospect, especially for the new subsurface observations. In addition, as this is a data description paper, I recommend that the authors share their code for data processing. At present, the manuscript reads more like a research article than a data paper.
**Answer:** Thanks for your comments. We had revised the related content in the subsection in "2.2 Field observations". Since the data collected by Argo, glider, drifter, RBRmaestro3, and UVP were automatically processed, this manuscript did not introduce processing procedures in details. The code for eddy detection is private since its copyright belongs to the Science and Technology Innovation Project for Laoshan Laboratory (Grant Nos. LSKJ202201406 and LSKJ202204303). The other code in Python for data processing and analysis is now available at https://github.com/Yezi-Yezi/Codes-for-the-Integrated-Observation-of-an-Asymmetric-Eddy-Dipole-in-SCS.

Other comments:

**4. Questions and Comments:** Line 101 The half-power wavelength cutoffs of 20° in longitude and 10° in latitude appear excessively large. For comparison, Pegliasco et al. (2022) set this value to 700 km.

**Answer:** Thank you for pointing this out. Figure 1 displays the eddy identification in the study area using the half-power wavelength cutoffs of 20° in longitude and 10° in latitude (Gaussian filter) as well as 700 km (Lanczos filter). As we can see, different filters impact the identification of eddy core and boundary. However, the AE and CE studied here, as larger black stars show, change very little under both cutoffs. We further provide the evolution of eddy kinematic properties in April 2023 (Figure 2). Similarly, although the time series of eddy properties displays modest fluctuations under two cutoffs, their variation exhibits a consistently coherent trend.

[Figure]

Figure 1. Eddy distribution in April 2023 under two half-power wavelength cutoffs.

[Figure]

Figure 2. The temporal evolution of (a) rotational speed, (b) amplitude, (c) EKE, and (d) radius for the dipole AE and CE under two filters. The diamonds mark the continuous coupling process from 13 to 22 April.

**5. Questions and Comments:** Line 171 T/S anomalies are calculated using monthly WOA dataset?
**Answer:** Thank you for pointing this out. The annual statistics of T/S profiles of WOA23 were used to compute the T/S anomalies. The annual and observed T/S profiles were resampled at the same depth through linear interpolation. The T/S anomalies were then obtained by subtracting climatological T/S profiles from the observed T/S profiles. The related text was added in Line 175-180 as "This work utilized the annual statistics of WOA23 dataset with a grid size of 1/4° to characterize the typical northern SCS water and Kuroshio water and calculate the T/S anomalies from the observations of Argo floats, gliders, and RBRmaestro3. The annual and observed T/S profiles were first resampled at the same depth through linear interpolation. The T/S anomalies were then obtained by subtracting annual T/S profiles from the observed T/S profiles."

**6. Questions and Comments:** In Figures 8, 9, and 10, it would be better to change the color of the missing values, since white is already used in the colormap.
**Answer:** Thank you for your helpful comment. We have revised the figures (Fig.8-Fig.12) as suggested. The gray areas now represent missing values in the below figures.

[Figure]

**Figure 8: T/S (black lines) and corresponding anomaly (colored shadow) observed by the Argo floats. The gray in each plot indicates missing values. The Video S2 presents the positions of Argo floats and the daily distribution of T/S profiles in detail.**

[Figure]

**Figure 9: T/S (black lines) and corresponding anomaly (colored shadow) observed by the gliders. The gray in each plot indicates missing values. (a) A diagram of dipole coordinate. The distance is the spatial separation between the CE center and the gliders. The azimuth angle $\theta$ is defined as the angle of the ray from the CE center to the glider relative to the ray through dipole eddy centers. The Video S3 presents the glider positions and the daily distribution of T/S profiles in detail.**

[Figure]

**Figure 10: (a) Temperature (black lines) and corresponding temperature anomaly (colored shadow), and (b) salinity (black lines) and corresponding salinity anomaly (colored shadow) crossing the dipole eddy centers. The profiles between two dashed blue (red) lines indicate stations are located within the CE (AE), and the profiles between a dashed red line and a dashed blue line indicate stations are located at the contact zone. The gray in each plot indicates missing values.**

[Figure]

**Figure 11: (a) Chl *a* concentration and (b) dissolved O₂ saturation crossing the dipole eddy centers. The dashed gray lines in (a) are the dissolved oxygen (O₂) saturation. The dashed red and blue lines have the same meaning as Fig.8. The gray in each plot indicates missing values. It is noted that the 2.0 μg/L of the color bar is the value exceeding 2.0 μg/L.**

[Figure]

**Figure 12: The particle concentrations of (a) microorganisms, (b) small zooplankton, and (c) large zooplankton crossing the dipole eddy centers. The dashed red and blue lines have the same meaning as Fig.8. The gray in each plot indicates missing values. Blanks in (c) mean there is no large zooplankton.**

**7. Questions and Comments:** In Figure 12, I am surprised that diel vertical migration of zooplankton was not observed. It may be more appropriate to plot the results on a logarithmic scale.

Guidi, L., Calil, P. H., Duhamel, S., Björkman, K. M., Doney, S. C., Jackson, G. A., ... & Karl, D. M. (2012). Does eddy-eddy interaction control surface phytoplankton distribution and carbon export in the North Pacific Subtropical Gyre?. Journal of Geophysical Research: Biogeosciences, 117(G2).

**Answer:** Thanks for your useful suggestions. We have revised the Fig.12 as suggested. Now the diel vertical migration of zooplankton can be observed. The related text was added as "Furthermore, the observed eddies are found to exert a significant influence on the vertical distribution of small zooplankton. A secondary subsurface maximum of small zooplankton occurs at approximately 75 m depth in close proximity to the CE core. In contrast, abundances are significantly suppressed within the AE core, especially at the 80–160 m depth, but exhibit elevated values within the AE–CE transition zone. This finding is consistent with Guidi et al. (2012) that concentrations of buoyant particles increase within the mesoscale frontal zone of AE-CE dipole." Please see Line 405-410 in the revised manuscript.

[Figure]

**Figure 12: The particle concentrations of (a) microorganisms, (b) small zooplankton, and (c) large zooplankton crossing the dipole eddy centers. The dashed red and blue lines have the same meaning as Fig.8. The gray in each plot indicates missing values. Blanks in (c) mean there is no large zooplankton.**

---

## Author Comment (AC3)

<Journal Name> Earth System Science Data

< MS No.> essd-2025-276

<MS type> Data description paper

<Manuscript Title> Integrated Observation of an Asymmetric Eddy Dipole in the South China Sea

**Dear Editors and Reviewers,**

We highly appreciate the detailed and valuable comments of the referees on our manuscript entitled "Integrated Observation of an Asymmetric Eddy Dipole in the South China Sea (MS No. essd-2025-276)". These comments are all valuable and helpful for revising and improving our paper, as well as providing important guidance for our research. In the past few days, we have referred to the comments and improved the paper.

As follows, we would like to clarify some of the points raised by the Associate Editor and Reviewers. The original comments begin with "**Questions and Comments**" and are quoted in normal font, the replies are in blue letters, the revised sentences and phrases are in red letters, and the line number in the revised manuscript is highlighted in yellow. We appreciate the Editors/Reviewers' warm work and taking the time to review the manuscript, and we hope that the corrections will meet with approval.

Yours Sincerely,

Shuang Long, Fenglin Tian*, Junwu Tang, Fangjie Yu, Fang Zhang, Wei Ma, Xinglong Zhang, Ge Chen*

2025-12-01

**1. Questions and Comments:** This manuscript integrates multiple observations from satellite altimeters, Argo floats, gliders, drifters, and survey vessels to investigate a case of an asymmetric eddy dipole in the South China Sea, with a particular focus on the evolution of its vertical structure. However, two reasons prevent me from recommending it for further consideration in Earth System Science Data. **Answer:** Thanks for your comments.

**2. Questions and Comments:** (1) I cannot identify sufficient novel aspects of this work to match the high impact expected for ESSD. Although the authors provide a dataset collected from multiple observation platforms, it is limited to a single local case of an asymmetric eddy dipole. Therefore, I do not believe this dataset has the potential to attract sufficient interest from the international research community. As noted in the authors' review in the introduction, many previous studies have revealed the vertical structure of asymmetric eddy dipoles using observational data (Line 50-63). In addition, the results of this manuscript mainly offer a simple descriptive analysis of the observations, without presenting any new findings that distinguish this study from previous work. The "gear-like" process has already been proposed in their previous studies based on global analyses. The vertical structures of temperature, salinity, dissolved oxygen, chlorophyll, and zooplankton have already been well documented in numerous previous studies. Although I acknowledge that most of those studies focused on anticyclonic and cyclonic eddies rather than asymmetric eddy dipoles, it remains unclear what new findings this work provides that distinguish it from previous research using the same type of dataset. The authors claim that their observations provide evidence of the impacts of asymmetric eddy dipoles on the vertical transport of water. However, this point has already been well demonstrated by previous numerical and observational studies, such as Guidi et al. (2012).
**Answer:** Thanks for your comments. We totally agree that many previous studies analyzed the vertical structure or explored the impact of eddy-eddy interaction on ecosystem of asymmetric eddy dipoles. Our previous work revealed the "gear-like" process of asymmetric dipole on the global scale, but it is primarily based on the eddy surface feature. The underwater structure and corresponding evolution of asymmetric dipole on global scale are a challenging task due to a lack of enough underwater information. Therefore, more *in-situ* data regarding asymmetric dipole are needed, and this manuscript provides another typical coupling process in SCS. In addition, this manuscript focuses on the response between changes in surface feature of dipole eddies and changes in vertical structure via eddy-eddy interaction. Since CE keeps steady and AE changes a lot during the coupling process, we concentrated on AE's vertical structure and found it is influenced by vertical transport of water.

We clarify this in t==he "Abstract"== that "Furthermore, the coupling interaction increased CE's positive temperature anomaly near the transitional zone and deepened AE's temperature and dissolved oxygen saturation structures, corresponding the strengthening of AE on 16-17 April. Both thermohaline and biological responses provide evidence that the interaction between the asymmetric dipole eddies impacted the vertical transport of water.", ==in the "Summary and discussion"== that "This work first confirmed the observed dipole eddies exhibit "gear-like" process during their first coupling process according to their kinematic properties and eddy shape derived from satellite altimetry data. A pair of AE and CE were generated in the west of the Luzon Strait and subsequently propagated together towards southwest. A 10−day successive coupling process from 13 to 22 April 2023 was identified. The kinematic characteristics of dipole eddies evidence that the stronger CE made the weaker AE strengthen as a result of the coupling effect, which similar with the "gear-like" process. However, the CE failed to prevent AE from weakening. Additionally, the eddy current between the dipole eddies flowed too fast to make the gliders not traverse the dipole eddies, requiring larger power gliders while they aim to transect the dipole eddies.
Subsequently, we further analyzed the evolution of vertical structure of dipole eddies from floats and survey vessel observations and found that the vertical structure of the asymmetric dipole eddies responses to their changes in surface feature. A conical shape of temperature anomaly ranging from

60 m to 350 m was observed within the AE, with a maximum of 3.2 ºC centered at around 150 m. Surprisingly, positive temperature anomalies (~0.5 ºC) at 50−300 m was found near the CE periphery. Chl *a* concentration at 50−100 m exceeded 0.8 $\mu g/L$. The 63% and 80% contours of dissolved $O_2$ saturation roughly corresponded to the SCM lower boundary within the CE and AE, respectively. The contours of temperature, salinity, and dissolved $O_2$ saturation inside the AE deepened during the latter half of the coupling process under the influence of the coupling effect, indicating well thermohaline and biological responses to the surface "gear-like" process. These observations illustrate that the interaction between dipole eddies influences the vertical water transport."

**3 Questions and Comments:** (2) As a manuscript submitted as a data description paper, I believe the current style and format do not provide sufficient details about the dataset. I recommend that the authors pay more attention to thoroughly describing the data collection methods, sensors used, processing procedures, temporal resolution, instructions, and application prospect, especially for the new subsurface observations. In addition, as this is a data description paper, I recommend that the authors share their code for data processing. At present, the manuscript reads more like a research article than a data paper.

**Answer:** Thanks for your comments. We had revised the related content in the subsection in "2.2 Field observations". Since the data collected by Argo, glider, drifter, RBRmaestro3, and UVP were automatically processed, this manuscript did not introduce processing procedures in details. The code for eddy detection is private since its copyright belongs to the Science and Technology Innovation Project for Laoshan Laboratory (Grant Nos. LSKJ202201406 and LSKJ202204303). The other code in Python for data processing and analysis is now available at https://github.com/Yezi-Yezi/Codes-for-the-Integrated-Observation-of-an-Asymmetric-Eddy-Dipole-in-SCS.

**2.2 Field observations**

Eddies in the SCS were detected and monitored for several months in advance utilizing near-real-time SLA data. Ultimately, an eddy pair in the western Luzon Strait was selected for observation (Fig.1 and Fig.2). During the research cruise taking place from April 10 to April 25, 2023, the position of the target eddy pair was first detected each day following the method in Sect. 2.1, and its next position was forecasted using two different algorithms (Ge et al., 2023; Chen et al., 2024). The results were then communicated to the survey vessel to guide the cruise and float deployment.

This research cruise was designed to gain the hydrographic and biological sections crossing the eddy centers of the CE-AE pair using a RBRmaestro3 multi-channel logger and an Underwater Vision Profiler (UVP). The survey vessel was underway from Zhanjiang, Guangdong Province on 10 April 2023 and successfully reached the target CE after two days (12 April 2023). Subsequently, the vessel arrived at the AE center on 15 April 2023 and turned back to CE on 17 April 2023. The vessel exited the target CE on 18 April 2023 and returned to Zhanjiang on 25 April 2023. Throughout the cruise, a total of twenty-nine stations were surveyed, with seventeen consecutive stations traversing the dipole eddy centers from April 12 to April 18, 2023 (colorful dots in Fig.1(c)). Table 1 details the seventeen consecutive stations.

**Table 1. Details of the seventeen consecutive stations regarding to dipoles.**

| station | time | longitude | latitude |
|---|---|---|---|
| 1 | 2023/04/12 08:00 | 18° 31.2N | 114° 33.0E |
| 2 | 2023/04/12 14:00 | 18° 50.028 N | 114° 53.022 E |
| 3 | 2023/04/13 06:00 | 18° 31.349 N | 114° 28.465 E |
| 4 | 2023/04/13 12:00 | 18° 28.30 N | 114° 22.72 E |
| 5 | 2023/04/13 18:00 | 18° 26.361 N | 114° 19.036 E |
| 6 | 2023/04/14 00:00 | 18° 24.971 N | 114° 17.195 E |
| 7 | 2023/04/14 14:30 | 18° 48.065 N | 114° 62.829 E |
| 8 | 2023/04/15 07:30 | 19° 11.811 N | 115° 04.988 E |
| 9 | 2023/04/15 14:00 | 19° 34.776 N | 115° 19.364 E |
| 10 | 2023/04/16 06:00 | 19° 52.052 N | 115° 31.989 E |
| 11 | 2023/04/16 12:00 | 19° 49.063 N | 115° 29.455 E |
| 12 | 2023/04/16 18:00 | 19° 45.648 N | 115° 25.897 E |
| 13 | 2023/04/16 23:30 | 19° 44.783 N | 115° 21.304 E |
| 14 | 2023/04/17 06:00 | 19° 46.941 N | 115° 14.974 E |
| 15 | 2023/04/17 12:00 | 19° 28.427 N | 115° 17.057 E |
| 16 | 2023/04/17 18:00 | 18° 54.822 N | 115° 00.028 E |
| 17 | 2023/04/18 06:00 | 18° 35.512 N | 114° 37.284 E |
| 18 | 2023/04/18 12:30 | 18° 20.346 N | 114° 20.156 E |

RBRmaestro3 multi-channel logger supports up to ten sensors on a single platform, allowing for a diversity of sensor configurations that can be fine-tuned for various applications. Its substantial storage capacity and reliable battery power enable extended deployments with higher sampling rates, making it particularly suitable for in situ observations. At each station, the RBRmaestro3 multi-channel instrument collected data every 0.5 seconds from the ocean surface to ~350 m depth, including conductivity, temperature, salinity, oxygen, chlorophyll a concentration (Chl a), and so on. Ruskin is the official software tool to operate RBRmaestro3 and is able to export recorded data after calibration. UVP serves as a low-power, cost-effective, and deep-ocean-rated in situ camera designed for automatically monitoring particles and plankton from autonomous platforms. It is capable of counting and sizing large particles with an equivalent spherical diameter (ESD) exceeding 100 $\mu m$. The UVP captured images of plankton and marine snow at a 5 m interval from surface to ~650 m depth. Benefiting from a full software ecosystem, such as Zooprocess, Ecotaxa, and EcoPart, images were first performed light correction, then segmentation, and finally counts for different sizes of particles. Drifter is an extremely valuable tool for reflecting near-surface ocean currents (Lumpkin et al., 2013). It is widely used to investigate the coherent mesoscale eddies that can trap Lagrangian particles over long periods of time (Lumpkin, 2016; Liu et al., 2016; Chen et al., 2021). To observe the geostrophic currents around the target eddies, two drifters were deployed in the CE and AE on April 12[th] and April 16[th], 2023, respectively (two black crosses in Fig.1(b)). Drifter_1 recorded data from April 12[th] to 24[th]

July and stayed within the CE until May 5[th], while Drifter_2 ran for 101 days from April 16[th] to August 3[rd] and was trapped in the AE until May 11[th].

The HM2000 Argo profile float is novel observation equipment and is able to automatically drift over a long time. After approbation by the International Argo Project, it has been utilized for the construction and maintenance of the Global Argo Real-time Ocean Observing Network (Zhang, 2018). Two hydrographic Argo floats were employed to daily record the T/S profiles inside the target AE and CE, with a maximum measure depth of 2000 m (two purple crosses in Fig.1(b)). One Argo float (No.898133) was trapped within the CE and recorded a total of 12 profiles from 12 to 16 April 2023. The other Argo float (No.897875) was deployed inside the AE on 16 April 2023 and finally ceased operation on 17 September 2023. It remained within the AE for only 8 days as the AE weakened.

The Petrel underwater glider, engineered by Tianjin University, is a buoyancy-driven and propeller-driven unmanned underwater vehicle that can continuously measure physical parameters (e.g., temperature and salinity) for a long period (Yang et al., 2019). In this work, Gilder_1 was deployed inside the CE on 12 April 2023, and Glider_2 was placed at the southwest outside of the CE on the same day (two green crosses in Fig.1(b)). As scheduled, the gliders should have traversed the dipole eddies to acquire high-resolution measurements from the surface down to 1000 m. However, dipole eddies coupled strongly, leading to the high-speed eddy current between them. As a consequence, the gliders lacked sufficient power to traverse the target dipole eddies like the cruise and ultimately remained within the dipole CE for a long time.

Other comments:
**4. Questions and Comments:** Line 101 The half-power wavelength cutoffs of 20° in longitude and 10° in latitude appear excessively large. For comparison, Pegliasco et al. (2022) set this value to 700 km.

**Answer:** Thank you for pointing this out. Figure 1 displays the eddy identification in the study area using the half-power wavelength cutoffs of 20° in longitude and 10° in latitude (Gaussian filter) as well as 700 km (Lanczos filter). As we can see, different filters impact the identification of eddy core and boundary. However, the AE and CE studied here, as larger black stars show, change very little under both cutoffs. We further provide the evolution of eddy kinematic properties in April 2023 (Figure 2). Similarly, although the time series of eddy properties displays modest fluctuations under two cutoffs, their variation exhibits a consistently coherent trend.

[Figure]

Figure 1. Eddy distribution in April 2023 under two half-power wavelength cutoffs.

[Figure]

Figure 2. The temporal evolution of (a) rotational speed, (b) amplitude, (c) EKE, and (d) radius for the dipole AE and CE under two filters. The diamonds mark the continuous coupling process from 13 to 22 April.

**5. Questions and Comments:** Line 171 T/S anomalies are calculated using monthly WOA dataset?
**Answer:** Thank you for pointing this out. The annual statistics of T/S profiles of WOA23 were used to compute the T/S anomalies. The annual and observed T/S profiles were resampled at the same depth through linear interpolation. The T/S anomalies were then obtained by subtracting climatological T/S profiles from the observed T/S profiles. The related text was added in Line 175-180 as "This work utilized the annual statistics of WOA23 dataset with a grid size of 1/4° to characterize the typical northern SCS water and Kuroshio water and calculate the T/S anomalies from the observations of Argo floats, gliders, and RBRmaestro3. The annual and observed T/S profiles were first resampled at the same depth through linear interpolation. The T/S anomalies were then obtained by subtracting annual T/S profiles from the observed T/S profiles."

**6. Questions and Comments:** In Figures 8, 9, and 10, it would be better to change the color of the missing values, since white is already used in the colormap.
**Answer:** Thank you for your helpful comment. We have revised the figures (Fig.8-Fig.12) as suggested. The gray areas now represent missing values in the below figures.

[Figure]

**Figure 8: T/S (black lines) and corresponding anomaly (colored shadow) observed by the Argo floats. The gray in each plot indicates missing values. The Video S2 presents the positions of Argo floats and the daily distribution of T/S profiles in detail.**

[Figure]

**Figure 9:** T/S (black lines) and corresponding anomaly (colored shadow) observed by the gliders. The gray in each plot indicates missing values. (a) A diagram of dipole coordinate. The distance is the spatial separation between the CE center and the gliders. The azimuth angle $\theta$ is defined as the angle of the ray from the CE center to the glider relative to the ray through dipole eddy centers. The Video S3 presents the glider positions and the daily distribution of T/S profiles in detail.

[Figure]

**Figure 10: (a) Temperature (black lines) and corresponding temperature anomaly (colored shadow), and (b) salinity (black lines) and corresponding salinity anomaly (colored shadow) crossing the dipole eddy centers. The profiles between two dashed blue (red) lines indicate stations are located within the CE (AE), and the profiles between a dashed red line and a dashed blue line indicate stations are located at the contact zone. The gray in each plot indicates missing values.**

[Figure]

**Figure 11: (a) Chl *a* concentration and (b) dissolved O₂ saturation crossing the dipole eddy centers. The dashed gray lines in (a) are the dissolved oxygen (O₂) saturation. The dashed red and blue lines have the same meaning as Fig.8. The gray in each plot indicates missing values. It is noted that the 2.0 μg/L of the color bar is the value exceeding 2.0 μg/L.**

[Figure]

**Figure 12: The particle concentrations of (a) microorganisms, (b) small zooplankton, and (c) large zooplankton crossing the dipole eddy centers. The dashed red and blue lines have the same meaning as Fig.8. The gray in each plot indicates missing values. Blanks in (c) mean there is no large zooplankton.**

**7. Questions and Comments:** In Figure 12, I am surprised that diel vertical migration of zooplankton was not observed. It may be more appropriate to plot the results on a logarithmic scale.

Guidi, L., Calil, P. H., Duhamel, S., Björkman, K. M., Doney, S. C., Jackson, G. A., ... & Karl, D. M. (2012). Does eddy-eddy interaction control surface phytoplankton distribution and carbon export in the North Pacific Subtropical Gyre?. Journal of Geophysical Research: Biogeosciences, 117(G2).

**Answer:** Thanks for your useful suggestions. We have revised the Fig.12 as suggested. Now the diel vertical migration of zooplankton can be observed. The related text was added as "Furthermore, the observed eddies are found to exert a significant influence on the vertical distribution of small zooplankton. A secondary subsurface maximum of small zooplankton occurs at approximately 75 m depth in close proximity to the CE core. In contrast, abundances are significantly suppressed within the AE core, especially at the 80–160 m depth, but exhibit elevated values within the AE–CE transition zone. This finding is consistent with Guidi et al. (2012) that concentrations of buoyant particles increase within the mesoscale frontal zone of AE-CE dipole." Please see Line 405-410 in the revised manuscript.

[Figure]

**Figure 12: The particle concentrations of (a) microorganisms, (b) small zooplankton, and (c) large zooplankton crossing the dipole eddy centers. The dashed red and blue lines have the same meaning as Fig.8. The gray in each plot indicates missing values. Blanks in (c) mean there is no large zooplankton.**